# CryoEM of endogenous mammalian V-ATPase interacting with the TLDc protein mEAK-7

Yong Zi Tan[1], Yazan M Abbas[1], Jing Ze Wu[2,3], Di Wu[4,5], Kristine A Keon[1], Geoffrey G Hesketh[6], Stephanie A Bueler[1], Anne-Claude Gingras[6,7], Carol V Robinson[4,5], Sergio Grinstein[2,3], John L Rubinstein[1,3,8]

**V-ATPases are rotary proton pumps that serve as signaling hubs with numerous protein binding partners. CryoEM with exhaustive focused classification allowed detection of endogenous proteins associated with porcine kidney V-ATPase. An extra C subunit was found in ~3% of complexes, whereas ~1.6% of complexes bound mEAK-7, a protein with proposed roles in dauer formation in nematodes and mTOR signaling in mammals. High-resolution cryoEM of porcine kidney V-ATPase with recombinant mEAK-7 showed that mEAK-7's TLDc domain interacts with V-ATPase's stator, whereas its C-terminal α helix binds V-ATPase's rotor. This crosslink would be expected to inhibit rotary catalysis. However, unlike the yeast TLDc protein Oxr1p, exogenous mEAK-7 does not inhibit V-ATPase and mEAK-7 overexpression in cells does not alter lysosomal or phagosomal pH. Instead, cryoEM suggests that the mEAK-7:V-ATPase interaction is disrupted by ATP-induced rotation of the rotor. Comparison of Oxr1p and mEAK-7 binding explains this difference. These results show that V-ATPase binding by TLDc domain proteins can lead to effects ranging from strong inhibition to formation of labile interactions that are sensitive to the enzyme's activity.**

## Introduction

Vacuolar-type adenosine triphosphatases (V-ATPases) are large, membrane-embedded protein complexes that function as proton pumps in eukaryotic cells. V-ATPase activity is essential for acidification of numerous intracellular compartments including endosomes, lysosomes, and secretory vesicles (Forgac, 2007). In some specialized cells, V-ATPases pump protons from the cytoplasm to the extracellular environment, enabling activities ranging from acidification of the distal renal tubule lumen by kidney intercalated cells to dissolution of bone minerals by osteoclasts (Eaton et al,

2021b). The enzyme consists of a soluble catalytic $V_1$ region that hydrolyzes ATP and a membrane-embedded $V_O$ region responsible for proton pumping. ATP hydrolysis in the $A_3B_3$ subcomplex of the $V_1$ region induces rotation of the rotor subcomplex, which contains subunits D, F, d, and a ring of membrane-embedded c subunits (Vasanthakumar & Rubinstein, 2020). In the mammalian $V_O$ region, the c-ring comprises nine c subunits and a single c″ subunit, with subunits ATP6AP1/Ac45 and ATP6AP2/PRR trapped inside the ring (Abbas et al, 2020). Rotation of the c-ring against subunit a drives proton translocation through the membrane. Three peripheral stalks, each consisting of heterodimers of subunits E and G, hold subunit a, along with subunits e and f (known as RNAseK in mammals) (Abbas et al, 2020), stationary relative to the rotor. Proton pumping by V-ATPase is regulated by the reversible dissociation of the $V_1$ and $V_O$ regions (Kane, 1995; Sumner et al, 1995), with subunit H inhibiting ATP hydrolysis in the detached $V_1$ region (Parra et al, 2000) and subunit C separating from $V_1$ following dissociation (Tabke et al, 2014).

V-ATPase–driven acidification is necessary for targeting and post-translational modification of proteins in the Golgi (Kellokumpu, 2019), degradation of material in lysosomes (Mindell, 2012), and the uptake of cargo for secretory vesicles including synaptic vesicles (Hnasko & Edwards, 2012). V-ATPases targeted to the plasma membrane of specialized cells are responsible for acidifying the extracellular environment (Qin et al, 2012; Breton & Brown, 2013; Stransky et al, 2016). Because of their their central role in acidification of intracellular compartments in all cells and acidification of the extracellular environment in specialized cells, complete disruption of V-ATPase activity is embryonic lethal in mammals (Sun-Wada et al, 2000), whereas aberrant activity or expression is associated with several diseases. For example, defects in V-ATPase–mediated lysosomal degradation is linked to neurodegenerative diseases (Colacurcio & Nixon, 2016) with tissue-specific mutations associated with osteopetrosis (Kornak et al, 2000), and distal renal tubular acidosis (Karet et al, 1999; Smith et al, 2000).

[1]Molecular Medicine Program, The Hospital for Sick Children Research Institute, Toronto, Canada    [2]Program in Cell Biology, The Hospital for Sick Children, Toronto, Canada    [3]Department of Biochemistry, University of Toronto, Toronto, Canada    [4]Physical and Theoretical Chemistry Laboratory, University of Oxford, Oxford, UK    [5]Kavli Institute for Nanoscience Discovery, University of Oxford, Oxford, UK    [6]Lunenfeld-Tanenbaum Research Institute, Sinai Health System, Toronto, Canada    [7]Department of Molecular Genetics, University of Toronto, Toronto, Canada    [8]Department of Medical Biophysics, University of Toronto, Toronto, Canada

Correspondence: john.rubinstein@utoronto.ca
Yong Zi Tan's present address is Department of Biological Sciences, National University of Singapore, Singapore, Singapore.
Yong Zi Tan's present address is Disease Intervention Technology Laboratory, Immunos, Agency for Science, Technology and Research (A*STAR), Singapore, Singapore.

V-ATPases interact with other proteins in the cell (Lee et al, 1999; Lu et al, 2001). In particular, V-ATPases are increasingly recognized as having important roles in signaling pathways such as Wnt (Cruciat et al, 2010), mechanistic target of rapamycin (mTOR) (Zoncu et al, 2011), and Notch (Yan et al, 2009; Vaccari et al, 2010), with each pathway involving different proteins interacting with the enzyme. The numerous roles and interactions involving V-ATPases suggest that in cells the enzyme may be found in complex with many different binding partners.

Analysis of V-ATPase from kidney tissue identified ARHGEF7, DMXL1, EZR, NCOA7, OXR1, RPS6KA3, SNX27, and nine subunits of the CCT complex as V-ATPase associated proteins (Merkulova et al, 2015). Two of these proteins, NCOA7 and OXR1, contain TLDc domains (Tre2/Bub2/Cdc16 lysin motif domain catalytic), which has been proposed as a V-ATPase interacting module (Eaton et al, 2021a). These proteins are believed to protect against oxidative stress through an unknown mechanism (Finelli et al, 2016).

In yeast, the small soluble protein Oxr1p consists primarily of a TLDc domain alone and was recently found to promote disassembly of the yeast V-ATPase $V_1$ and $V_O$ regions (Khan et al, 2022). In contrast, numerous mammalian TLDc domain-containing proteins also possess additional domains (Finelli & Oliver, 2017). These include a polysaccharide-binding Lysine Motif (LysM) and protein- or lipid-binding GRAM domain in NCOA7 and OXR1, a TBC domain similar to those found in Rab-GTPase-activating proteins in TBC1D24, and a myristoylation motif and apparent calcium-binding EF-hand motif in mEAK-7. The biological functions of these mammalian TLDc proteins remain obscure, but mutations of the genes encoding them can cause diseases, such as a range of neurological disorders upon mutation of the gene for TBC1D24 (Finelli et al, 2019).

To identify low abundance complexes between mammalian V-ATPase and its binding partners, we first determined the subunit composition and structure of V-ATPase prepared from porcine kidney. We then subjected images of this complex to an exhaustive 3D classification strategy designed to detect low-population structures. This procedure revealed a small population of V-ATPase complexes with super-stoichiometric occupancy of the C subunit, and a second small population of complexes with an additional density bound to the catalytic $A_3B_3$ subcomplex and one of the peripheral stalks. We tentatively identified this protein as the TLDc domain containing protein mEAK-7 using mass spectrometry and subsequently confirmed the interaction by determining a high-resolution structure of V-ATPase with recombinant mEAK-7 bound. Cells with mEAK-7 knocked down display abnormal mTOR signaling (Nguyen et al, 2018) and an mEAK-7 knockout mouse displays abnormal epididymis, lung, and skin morphology (https://www.mousephenotype.org/data/genes/MGI:1921597).

mEAK-7 has been proposed to activate the mTOR pathway but was not previously suggested to interact with V-ATPase (Mendonça et al, 2020; Nguyen et al, 2018, 2019). The structure shows that the TLDc domain and a C-terminal domain are critical for V-ATPase interaction. The mEAK-7 C-terminal domain is unique among TLDc-containing proteins and forms an α helix that binds to the central rotor of the V-ATPase, thereby crosslinking the stator and rotor parts of the enzyme. This crosslink would be expected to block rotation and ATP hydrolysis by the enzyme. Intriguingly, and in sharp contrast with the yeast TLDc protein Oxr1p (Khan et al, 2022),

mEAK-7 binding does not inhibit activity both in vitro and in cells, with ATP hydrolysis partially disrupting the interaction. This behavior suggests that mEAK-7 interaction with V-ATPase is sensitive to the activity of the enzyme, which may relate to mEAK-7's previously proposed roles in cellular signaling. Together, these results suggest a surprising diversity in the consequences and biological roles of TLDc domain protein interactions with V-ATPases.

# Results

## Overall structure of the porcine kidney V-ATPase

To identify low-abundance complexes between V-ATPases and the proteins that bind to it in cells, we isolated intact V-ATPase from porcine kidney using a fragment of the bacterial effector protein SidK (Abbas et al, 2020) (Fig 1A). Extracting the enzyme from membranes with the detergent glyco-diosgenin (GDN) retained subunit H in the complex, similar to extraction with other mild detergents (Wang et al, 2020a, 2020b). Multiple V-ATPase subunits have isoforms encoded by different genes that are expressed in a tissue-dependent manner and targeted to different intracellular compartments (Toei et al, 2010). Mass spectrometry identified the isoform composition of the porcine kidney V-ATPase preparation as subunits A, B1, C, D, E1, F, G1, and H in the $V_1$ region, and subunits a1, c, c″, d1, e2, RNAseK, ATP6AP1/Ac45, and ATP6AP2/PRR in the $V_O$ region (Figs 1A and S1 and Tables S1 and S2). Notably, these subunit isoforms correspond to the ubiquitous lysosomal V-ATPase subunits and, other than subunit B1, do not include the kidney-specific isoforms C2, G3, a4, or d2 (Toei et al, 2010). Although initially surprising, this finding is consistent with the kidney-specific isoforms being expressed primarily in α-intercalated cells, which comprise <4% of kidney cells (Park et al, 2018). Despite the presence of bound SidK, which inhibits V-ATPase by ~30% (Zhao et al, 2017; Maxson et al, 2021), the specific ATPase activity of the preparation was 2.9 ± 0.72 μmol ATP/min/mg (±s.d., n = 6 independent assays from two separate enzyme preparations). The preparation is also highly coupled, showing 91% inhibition following addition of 1 μM bafilomycin.

CryoEM of the preparation, combined with 3D classification during image analysis, led to three structures of the complex in each of its three main rotational states (Zhao et al, 2015) at nominal resolutions of 3.7–4.1 Å (Fig 1B, left and Figs S2 and S3 and Table 1). Focused refinement of these maps resolved the $V_1$ regions at 3.6–4.0 Å resolution and the $V_O$ regions at 3.7–5.8 Å resolution (Figs S2 and S3). Combined with the identities of subunit isoforms from mass spectrometry, the cryoEM maps allowed for construction of atomic models of the porcine kidney V-ATPase in the different rotational states (Fig 1B, right and Figs 1C and S4). These atomic models were complete except for residues 8–82 of subunit G1 in peripheral stalk 1 and residues 10–115 of subunit E1 in peripheral stalk 2, which were modelled as poly-alanine, and residues 667–713 of subunit a1, which could not be modelled. Overall, the models of the porcine kidney V-ATPase closely resemble previous mammalian V-ATPase structures (Wang et al, 2020a, 2020b; Abbas et al, 2020).

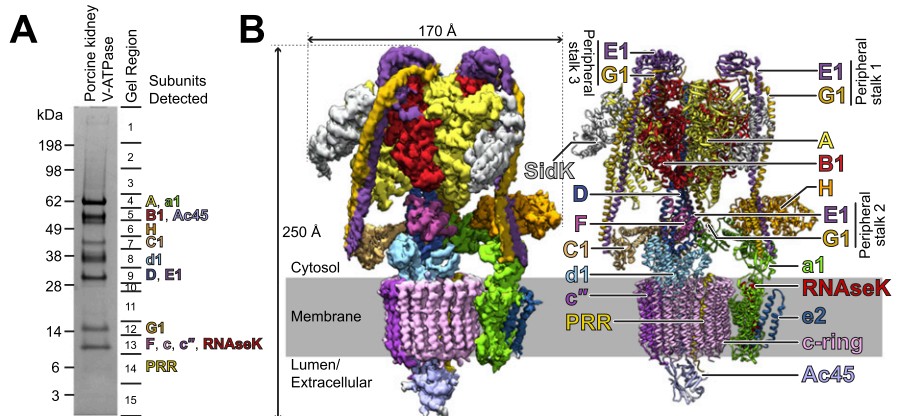

**Figure 1. Structure of porcine kidney V-ATPase.**
**(A)** SDS–PAGE gel showing subunit isoforms identified by mass spectrometry. **(B)** CryoEM map (*left*) and atomic model (*right*) of V-ATPase from porcine kidney. **(C)** Additional densities (*green*) bound to the V-ATPase. **(D)** Fitting of an atomic model for subunit C into one of the additional densities. **(E)** Fitting of a homology model for mEAK-7 into the other additional density.

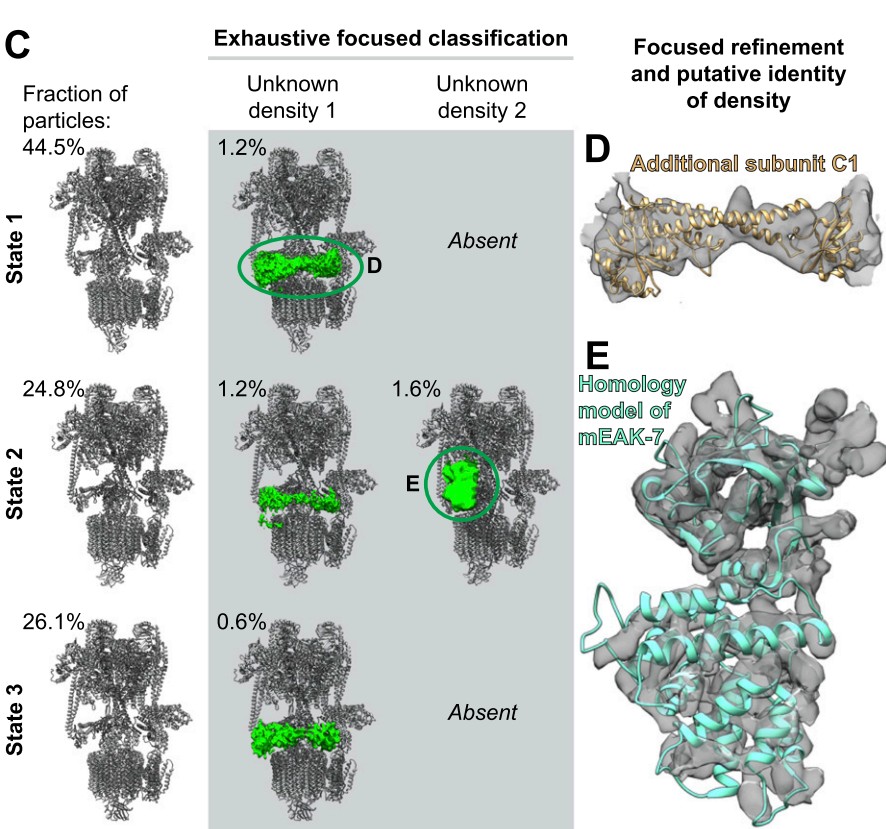

## Some kidney V-ATPase complexes have an additional C subunit or mEAK-7 bound

To search for endogenous sub-stoichiometric binding partners of V-ATPase, signal corresponding to the known parts of the complex was estimated from the atomic models and subtracted from images (Bai et al, 2015). 3D classification without refinement of particle orientation (Scheres et al, 2007) was then performed within masks adjacent to the complex that were designed to contain any density that is not part of the V-ATPase (Fig S5). To facilitate this analysis, a dataset comprising 737,357 particle images, including protein with a variety of nucleotide analogues, was subjected to the procedure. 3D

classification revealed two small populations of complexes, each with a different additional density bound to the V-ATPase (Fig 1C). The first population comprised ~3% of the dataset and is found in all three of the main rotational states of the enzyme, with 1.2% in State 1, 1.2% in State 2, and 0.6% in State 3. The additional density bridges peripheral stalks 1 and 3 of the $V_1$ region of the enzyme, which are usually not connected (Fig 2B or C, *middle column*). This density appears to correspond to an additional copy of subunit C (Fig 1D), which typically bridges peripheral stalks 2 and 3 (Fig 1B and D). The observation of a population of V-ATPase complexes with two copies of subunit C explains previous native mass spectrometry of rat brain V-ATPase that indicated the existence of complexes with two C subunits (Abbas et al,

**Table 1.   Cryo-EM data collection and modeling statistics for V-ATPase alone.**

| | State1 | State2 | State3 |
|---|---|---|---|
| EM data collection/processing | | | |
| Microscope | | FEI Titan Krios | |
| Voltage (kV) | | 300 | |
| Camera | | Falcon 4 | |
| Mode | | Counting | |
| Defocus mean ± std ($\mu m$) | | 1.7 ± 0.4 | |
| Exposure time (s) | | 9 | |
| Number of fractions | | 29 | |
| Exposure rate (e⁻/pixel/s) | | 5 | |
| Total exposure ($e^-/Å^2$) | | 40 | |
| Pixel size (Å) | | 1.01867 | |
| Number of micrographs | | 5,451 | |
| Number of particles (after cleanup) | | 133,337 | |
| Number of particles (in final map) | 24,327 | 14,746 | 22,866 |
| Symmetry | C1 | C1 | C1 |
| Resolution (global) (Å) | 3.7 | 4.1 | 3.8 |
| Directional resolution range (Å) | 3.5–3.9 | 3.9–5.0 | 3.6–4.2 |
| Sphericity of 3DFSC | 0.92 | 0.74 | 0.89 |
| SCF Value[a] | 0.84 | 0.84 | 0.84 |
| Map B factor ($Å^2$) | 76 | 70 | 77 |
| EMDB ID | EMD-26386 | EMD-26387 | EMD-26388 |
| Model statistics | | | |
| Residues | 8978 | 8978 | 8978 |
| Ligand | ADP | ADP | ADP |
| Map CC | 0.78 | 0.70 | 0.76 |
| RMSD [bonds] (Å) | 0.021 | 0.020 | 0.021 |
| RMSD [angles] (Å) | 1.724 | 1.802 | 1.777 |
| All-atom clashscore | 0.84 | 0.85 | 0.82 |
| Ramachandran plot | | | |
| Favored (%) | 97.74 | 97.68 | 97.76 |
| Allowed (%) | 2.01 | 2.06 | 2.02 |
| Outliers (%) | 0.25 | 0.26 | 0.21 |
| Rotamer outliers | 0.01 | 0.06 | 0.08 |
| C-$\beta$ deviations | 0.06 | 0.07 | 0.04 |
| MolProbity score | 0.82 | 0.83 | 0.81 |
| EM-Ringer score | 1.26 | 0.42 | 1.05 |
| PDB ID | 7U8P | 7U8Q | 7U8R |

[a]The sampling compensation factor (SCF) value is calculated as described (Baldwin & Lyumkis, 2020) and assumes that all orientations have been determined accurately.

2020). The significance of this variant of the structure is not clear, but it may correspond to V-ATPase complexes that misassemble because of the dynamic nature of the association of the $V_1$ and $V_O$ regions.

The second additional density is found in ~1.6% of particle images and only appeared with the complex in rotational State 2

(Fig 1C, *right column*). This density abuts subunit B, subunit A, and peripheral stalk 3. The shape of the density does not correspond to any of the known subunits of V-ATPase. It is not clear if the fraction of particle images with this additional density represents the true fraction of V-ATPase complexes in cells with this protein bound, or if

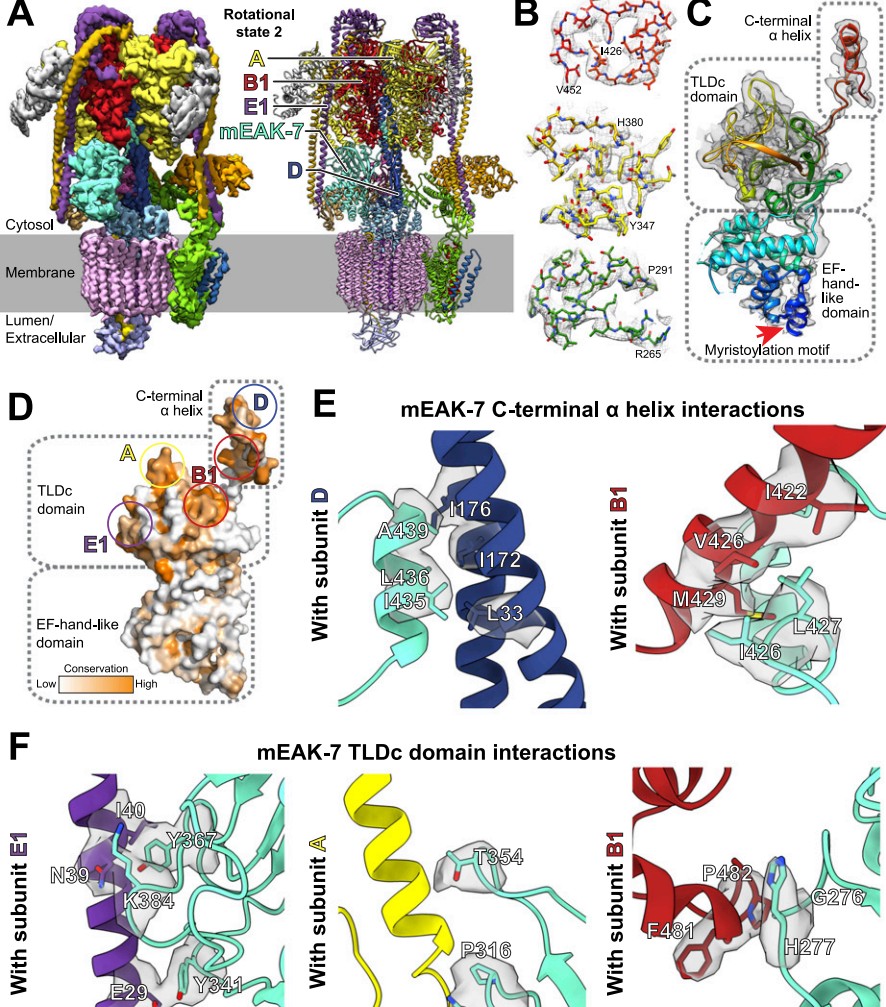

**Figure 2. Structure of mEAK-7 bound to V-ATPase crosslink's the enzyme's rotor and stator.**
**(A)** CryoEM map (*left*) and atomic model (*right*) for recombinant mEAK-7 bound to V-ATPase. **(B)** Examples of model in map fit for mEAK-7. **(C)** Domain structure of mEAK-7. The myristoylation motif in mEAK-7 is indicated (*red arrow*). **(D)** Sequence conservation in mEAK-7. **(E)** Interactions between the C-terminal α helix of mEAK-7 and the rotor of V-ATPase. **(F)** Interactions between the TLDc domain of mEAK-7 and subunits from the stator of V-ATPase.

the interaction is more common but is disrupted during purification of V-ATPase. The list of proteins identified by mass spectrometry of the preparation (Table S1) was inspected to identify candidate proteins that could explain the density. After excluding V-ATPase subunits, the most abundant ~50-kD proteins based on iBAQ scores (intensity-based absolute quantification) were the α and β subunits of mitochondrial ATP synthase followed by mEAK-7. Structures are known for both subunit α and β from ATP synthase (Abrahams et al, 1994), and they do not fit the unexplained density, excluding them as the bound protein in the low-resolution cryoEM map. In contrast, a homology model of mEAK-7 could be fit in the density with a clear agreement between α helices in the model and map density (Fig 1E), suggesting that this extra density could be due to mEAK-7.

### mEAK-7 interacts with catalytic subunits and a peripheral stalk from V-ATPase

mEAK-7 is the mammalian homologue of *Caenorhabditis elegans* EAK-7 (enhancer-of-akt-1-7), which regulates dauer formation in

nematodes (Lee et al, 2001; Lin et al, 2001; Alam et al, 2010). Mutations in *C. elegans* EAK-7 promote diapause and longevity (Alam et al, 2010). In mammals, mEAK-7 has been proposed to activate an alternative mTOR pathway, thereby regulating S6K2 activation and 4E-BP1 repression (Nguyen et al, 2018). To test the hypothesis that the additional density seen in Fig 1E is indeed mEAK-7, DNA sequence encoding the protein was cloned into a vector with an N-terminal 6× His tag, expressed heterologously in *Escherichia coli*, and purified to homogeneity. Purified mEAK-7 was mixed with porcine V-ATPase at ~20× molar excess and cryoEM specimens were prepared and subjected to structure determination. Particle images were again sorted into classes corresponding to the three main rotational states of the enzyme. The map of rotational State 2 showed V-ATPase with mEAK-7 bound (Fig 2A, *left*, and Fig S6A and B and Table 2). Of the 86,338 particle images in the dataset, 43,251 (23%) contributed to maps of V-ATPase in State 2, and all of these images (i.e., 23% of total particle images) had density for mEAK-7. This statistic is in sharp contrast to the occupancy of mEAK-7 from native source, where 24.8% of particle images were found in State 2

**Table 2.** Cryo-EM data collection and modeling statistics for V-ATPase bound to mEAK-7.

| | V-ATPase + mEAK7 (state 2) |
|---|---|
| EM data collection/processing | |
| Microscope | FEI Titan Krios |
| Voltage (kV) | 300 |
| Camera | Falcon 4 |
| Mode | Counting |
| Defocus mean ± std ($\mu$m) | 2.0 ± 0.4 |
| Exposure time (s) | 17 |
| Number of fractions | 29 |
| Exposure rate ($e^-$/pixel/s) | 4 |
| Total exposure ($e^-$/Å$^2$) | 36 |
| Pixel size (Å) | 1.3225 |
| Number of micrographs | 5,142 |
| Number of particles (after cleanup) | 186,338 |
| EM processing | |
| Number of particles (in final map) | 56,808 |
| Symmetry | C1 |
| Resolution (global) (Å) | 3.5 |
| Directional resolution range (Å) | 3.3–3.7 |
| Sphericity of 3DFSC | 0.97 |
| SCF Value[a] | 0.70 |
| Map B factor (Å$^2$) | 118 |
| EMDB ID | EMD-26385 |
| Model statistics | |
| Residues | 9348 |
| Ligand | — |
| Map CC | 0.82 |
| RMSD [bonds] (Å) | 0.016 |
| RMSD [angles] (Å) | 1.798 |
| All-atom clashscore | 2.33 |
| Ramachandran plot | |
| Favored (%) | 97.60 |
| Allowed (%) | 2.19 |
| Outliers (%) | 0.22 |
| Rotamer outliers | 0.61 |
| C-$\beta$ deviations | 0.17 |
| MolProbity score | 1.10 |
| EM-Ringer score | 1.64 |
| PDB ID | 7U8O |

[a]The sampling compensation factor (SCF) value is calculated as described (Baldwin & Lyumkis, 2020) and assumes that all orientations have been determined accurately.

but only 6.4% of them (1.6% of total particle images) had density for mEAK-7. The resolution of the map allowed for construction of an atomic model of the complex (Fig 2A, *right*, and Fig 2B). This atomic model of mEAK-7 could be overlaid with the map from the endogenous complex with high fidelity (Fig S6C), confirming the identity of the additional endogenous binding protein shown in Fig 1E as mEAK-7. mEAK-7 contains an N-terminal myristoylated motif (Fig 2C, *red arrow*), followed by a region resembling an EF-hand domain (Fig 2C, *cyan*), a TLDc domain (Fig 2C, *yellow*, *green*, and *orange*), and a C-terminal $\alpha$ helix that is separated from the rest of the protein by an extended linker region (Fig 2C, *red*) (Finelli & Oliver, 2017). The N terminus of mEAK-7 is poorly resolved in the map, suggesting that it is disordered in the detergent-solubilized protein.

mEAK-7 binding to V-ATPase is mediated through conserved regions in its TLDc domain and C-terminal $\alpha$ helix (Fig 2D). The C-terminal $\alpha$ helix binds subunit D from the central rotor of V-ATPase (Fig 2E, *left*) as well as a B1 subunit (Fig 2E, *right*), facilitated mainly by hydrophobic interactions. This $\alpha$ helix is formed from sequence in the protein that was previously proposed to bind mTOR (Nguyen et al, 2018). The structure shows that this sequence is held against the D subunit away from the cytoplasm and could not interact with mTOR, at least when mEAK-7 is bound to V-ATPase. Binding of mEAK-7 to rotational State 2 causes numerous subtle conformational changes throughout the complex (Fig S6D and Video 1), including widening of the subunit A-B interface where the C-terminal $\alpha$ helix binds, and a slight rotation of the central rotor in the direction of ATP hydrolysis. The structure of the TLDc domain resembles previously determined TLDc domain structures (Fig S6E).

The TLDc domain binds V-ATPase through interactions with subunits E1, A, and B1 (Fig 2F). Binding involving subunit E1 occurs through apparent stacking interactions between Tyr341 from mEAK-7 and Glu29 from subunit E1, electrostatic interaction between Lys384 from mEAK-7 and Asn39 from subunit E1, and hydrophobic interactions between Tyr367 from mEAK-7 and Ile40 from subunit E1 (Fig 2F, *left*). Binding involving subunit A occurs through apparent ionic interactions between Thr354 and Gln317 from mEAK-7 with backbone atoms from residues Arg553 and Thr557 from subunit A, respectively, and hydrophobic interactions between Pro316 from mEAK-7 and Ala559 from subunit A (Fig 2F, *middle*). Residues Gly276 and His277 from the TLDc domain of mEAK-7 interact with backbone atoms from Phe481 and Pro482 of subunit B1 (Fig 2F, *right*). Together the TLDc domain and C-terminal $\alpha$ helix form a pincer-like grip around B1 (Fig 2A). The residues in B1 and E1 identified as interacting with mEAK-7 are conserved in the B and E subunit isoforms, suggesting that mEAK-7 binding is isoform insensitive.

The domain homologous to an EF-hand motif in mEAK-7 consists of seven $\alpha$ helices (Figs 2C and S7A). EF-hand motifs can bind or release other proteins in response to calcium binding (Burgoyne & Haynes, 2012). However, inspection of the sequence of the apparent EF-hand motif in mEAK-7 shows that it is missing key conserved acidic residues needed for calcium binding (Fig S7B) and is poorly conserved compared with rest of mEAK-7 (Fig 2D). Furthermore, titration of mEAK-7 with calcium did not notably change its circular dichroism spectrum (Fig S7C) and cryoEM in the presence of calcium or the calcium chelators EDTA and EGTA did not change the structure of the mEAK6:V-ATPase complex (Fig S7D). Consequently,

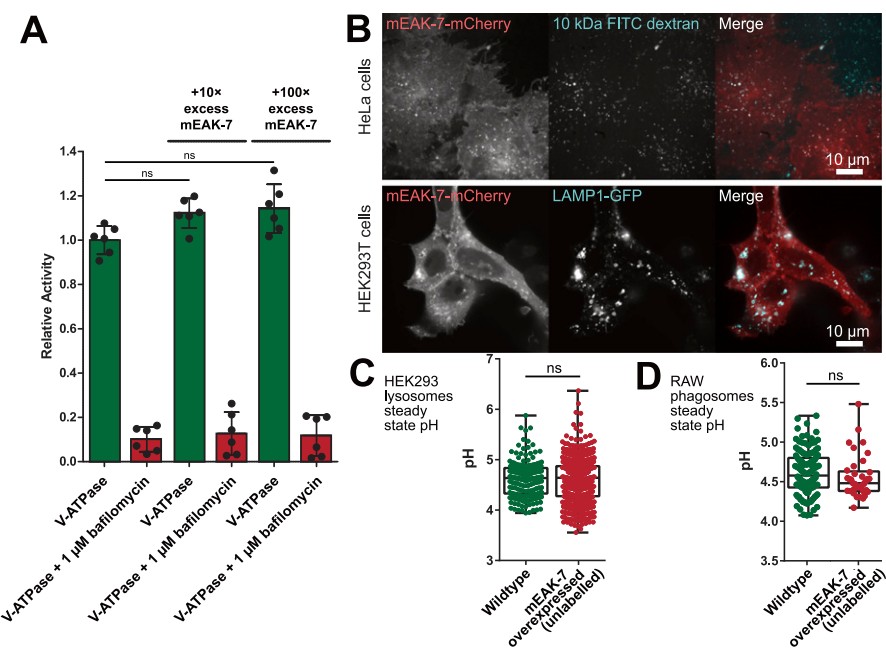

**Figure 3. mEAK-7 binding does not affect V-ATPase activity.**
**(A)** Recombinant mEAK-7 does not alter the ATPase activity or bafilomycin sensitivity of purified V-ATPase in vitro. **(B)** mEAK-7-mCherry localizes to lysosomes in HeLa cells (*upper*) and HEK293T cells (*lower*), which were labelled with fluorescein isothiocyanate (FITC)–conjugated 10-kD dextran and GFP-tagged LAMP1, respectively. **(C, D)** Overexpression of unlabelled mEAK-7 does not alter the steady state pH of lysosomes in HEK293T cells (C) or phagosomes in RAW 264.7 cells (D).

the apparent EF-hand–like motif in mEAK-7 appears to be structural rather than calcium sensing.

### mEAK-7 interaction with V-ATPase does not affect enzyme activity

mEAK-7 crosslinks the rotor and stator parts of V-ATPase, with its C-terminal $\alpha$ helix interacting with subunit D from the rotor and the TLDc domain interacting with peripheral stalk 3 and an AB pair (Fig 2A). This interaction would be expected to inhibit ATP hydrolysis and proton pumping by V-ATPase, similar to chemical crosslinks between the rotor and stator in other rotary ATPases (Srivastava et al, 2018). To assess how mEAK-7 affects V-ATPase activity, we performed ATPase assays with V-ATPase in the presence of recombinant mEAK-7 (Fig 3A). To our surprise, a 100× molar excess of mEAK-7 did not detectably inhibit ATPase activity. The excess of mEAK-7 also did not affect bafilomycin sensitivity of the preparation, showing that the $V_1$ and $V_O$ regions of the enzyme remained coupled in the presence of mEAK-7. This coupling shows that mEAK-7 does not cause dissociation of $V_1$ and $V_O$ with purified V-ATPase.

Perturbation of V-ATPase activity in cells would also be expected to affect the pH of lysosomes and phagosomes, which rely on V-ATPase for acidification. It is possible that mEAK-7 expression in cells modulates V-ATPase activity by influencing the assembly status of the $V_1$ region with the $V_O$ region (Kane, 1995; Sumner et al, 1995), which could affect the steady state pH of lysosomes and phagosomes. This hypothesis would be in line with the recent finding that yeast Oxr1p promotes dissociation of yeast V-ATPase (Khan et al, 2022). To detect mEAK-7–induced changes in V-ATPase activity in cells, we cloned mEAK-7 into a vector for overexpression in mammalian cells. Ectopic expression of mEAK-7 with a C-terminal mCherry tag in HeLa cells with lysosomes labelled with FITC-dextran (Canton & Grinstein, 2015) (Fig 3B, *upper*) confirmed the lysosomal localization of the protein, as shown previously (Nguyen

et al, 2018). Similarly, co-transfection of HEK293T cells mEAK-7 with a C-terminal mCherry tag and LAMP1 with a GFP tag showed lysosomal localization of mEAK-7 in HEK293T cells and confirmed successful expression in HEK293T cells (Fig 3B, *lower*). Endogenous mEAK-7 expression in HEK293T cells is nearly undetectable (Nguyen et al, 2018) so that any mEAK-7 in the cells is expected to be due to expression from the vector. As the C-terminal mCherry tag could interfere with binding of mEAK-7's C-terminal $\alpha$ helix to V-ATPase subunit D, subsequent experiments were performed with untagged mEAK-7 co-transfected with PLC$\delta$-PH-RFP, a PtdIns(4,5)P$_2$ biosensor that associated with the inner leaflet of the plasma membrane (Lemmon et al, 1995; Botelho et al, 2000) but not with lysosomes, to confirm successful transfection. Consistent with the in vitro ATPase assays, expression of this construct in HEK293T cells (Fig 3C) and macrophage-like RAW 264.7 cells (Fig 3D) did not lead to changes in the steady state lysosomal pH or phagosomal pH, respectively (Canton & Grinstein, 2015). Together, these experiments indicate that mEAK-7 expression neither inhibits nor activates V-ATPase in vitro or in cells. It is possible that porcine mEAK-7 expressed in human cells does not interact with human V-ATPase in precisely the same way as it does with porcine V-ATPase. However, it is worth noting that all the mEAK-7 residues identified above as interacting with V-ATPase subunits are conserved between porcine and human mEAK-7. Furthermore, it is also possible that mEAK-7 perturbation of lysosomal or phagosomal pH requires additional proteins that were not over-expressed here. mEAK-7 binding could even participate in V-ATPase assembly in some cellular contexts.

### V-ATPase binding by mEAK-7 involves its C-terminal $\alpha$ helix and is perturbed by ATP

To understand the relative importance of mEAK-7's TLDc domain and C-terminal $\alpha$ helix for V-ATPase binding, we prepared a

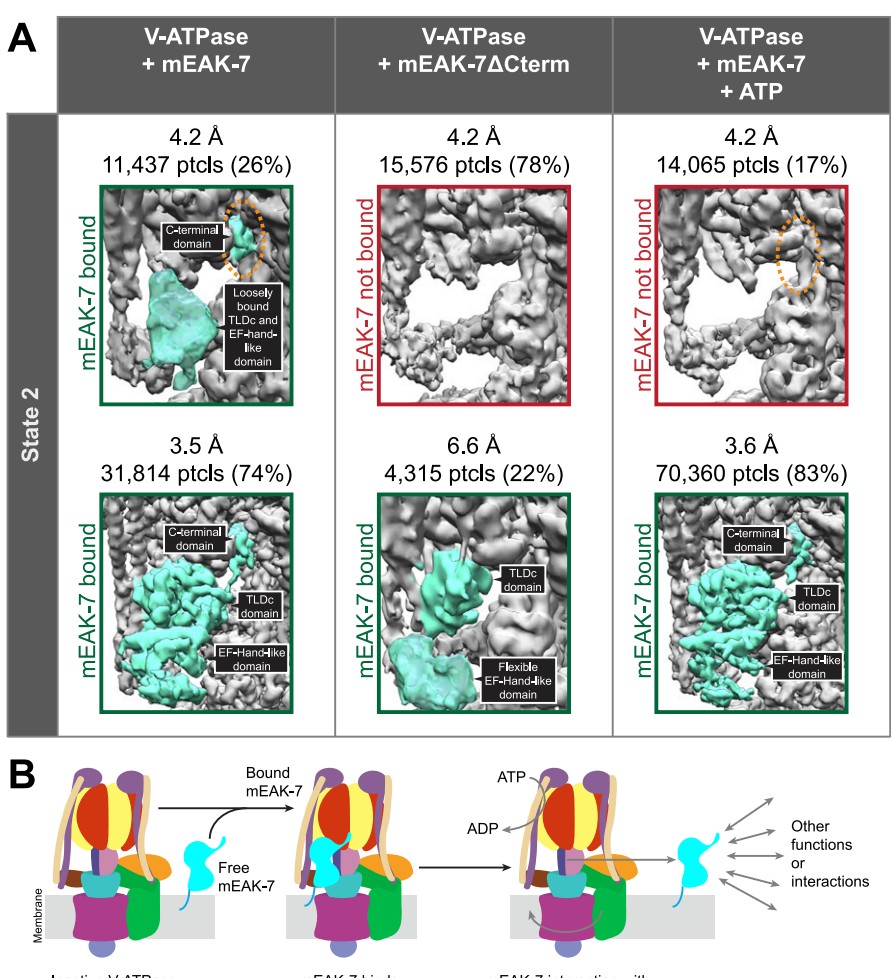

**Figure 4. C-terminal truncation and ATP hydrolysis suppress mEAK-7 binding to V-ATPase.**
**(A)** Intact mEAK-7 in the absence of ATP (*left*) binds V-ATPase in rotational State 2, with its C-terminal α helix well-resolved in all 3D classes (*dashed orange ellipse*). Truncation of the C-terminal α helix results in 3D classes lacking bound mEAK-7 (*center*). The addition of ATP to mEAK-7 and V-ATPase (*right*) results in a 3D class lacking density for the TLDc domain of mEAK-7 and its C-terminal α helix (*dashed orange ellipse*). **(B)** ATP hydrolysis induced disruption of the association between mEAK-7 and V-ATPase could enable alternative functions in the cell.

construct of mEAK-7 with the C-terminal α helix removed by truncation at residue Ser415. This construct was expressed heterologously in *E. coli*, purified to homogeneity, and mixed with V-ATPase at ~20× molar excess for structure determination by cryoEM. As described above, intact mEAK-7 binds V-ATPase in rotational State 2 and is seen in all 3D classes corresponding to this state (Fig 4A, *left*). Whereas 3D classification identified a population comprising ~25% of particle images where the density for mEAK-7's N-terminal domain and TLDc domain are weak and poorly resolved, suggesting that they are loosely attached via the TLDc domain, density for mEAK-7's C-terminal α helix remains strong and well resolved even in this 3D class (Fig 4A, *upper left*, *circled in orange*). In contrast, when the C-terminal α helix is truncated most V-ATPase complexes in rotational State 2 do not bind mEAK-7 (Fig 4A, *upper middle*). Only ~22% of particle images showed any density for the N-terminal domain and TLDc domain of mEAK-7 when the C-terminal α helix was truncated, and even this density was weak and poorly resolved (Fig 4A, *lower middle*). These experiments indicate that mEAK-7's C-terminal α helix is important for the tight binding of V-ATPase by mEAK-7.

The observation that mEAK-7, which crosslinks the rotor and stator of V-ATPase in the structure, does not affect the enzyme's activity in vitro or in cells suggests that the crosslink is broken during rotary catalysis. To determine the effect of ATP hydrolysis on mEAK-7 crosslinking of the rotor and the stator, V-ATPase was mixed with a ~20× molar excess of mEAK-7, ATP was added to 10 mM and mixed, and cryoEM grids were frozen within 5 s. Given the concentration of V-ATPase (20 mg/ml) and the enzyme's specific ATPase activity (2.9 ± 0.72 μmol ATP/min/mg), these conditions ensure that the grids were frozen before the supply of ATP was consumed. In the absence of ATP, it was impossible to find a 3D class in rotational State 2 that lacked density for mEAK-7 completely and density for mEAK-7's C-terminal α helix was always clear (Fig 4A, *upper left*). In contrast, with ATP added a population of particles appeared with mEAK-7 entirely absent in rotational State 2 (Fig 4A, *upper right*). In this state, even density for the C-terminal α helix is missing (Fig 4A, *upper right*, *circled in orange*). Thus, it appears that rotation of the rotor, driven by ATP hydrolysis can result in the displacement of mEAK-7 from V-ATPase and disruption of the crosslink between the enzyme's rotor and stator. Despite several biochemical experiments, we were unable to demonstrate that mEAK-7

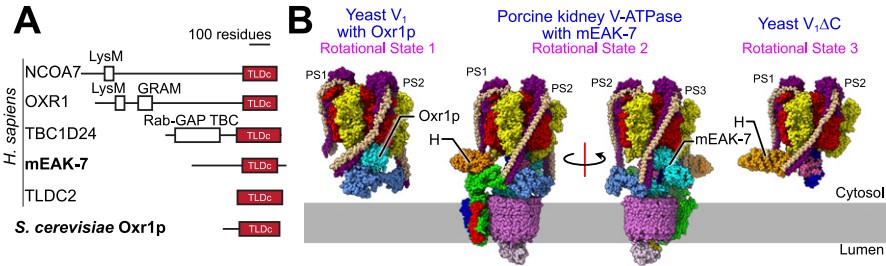

**Figure 5. TLDc domain interactions with V-ATPase.**
**(A)** TLDc domains are found in numerous different proteins. **(B)** The TLDc domain from the yeast protein Oxr1p (*left*) interacts with yeast $V_1$ complexes in rotational State 1, binds peripheral stalk 2 (PS2) and is mutually exclusive with subunit H binding. Oxr1p binding induces a sharp bend in PS2. mEAK-7 interacts with the intact V-ATPase in rotational State 2, binds peripheral stalk 3 (PS3), and does not affect subunit H binding (*middle two structures*). The structure is shown in two orientations to illustrate that mEAK-7 binding does not bend PS2 or PS3. A structure of yeast $V_1\Delta C$ (*right*) in rotational State 3 shows the sharp bend in PS2 induced by subunit H, which is reminiscent of the bend in PS2 induced by Oxr1p but not mEAK7.

can be fully dissociated from V-ATPase by the addition of ATP to the complex, including experiments where we washed immobilized V-ATPase:mEAK-7 extensively with a buffer containing ATP before elution of V-ATPase. These experiments suggest that mEAK-7 can remain associated to V-ATPase, either by its TLDc domain or its C-terminal $\alpha$ helix, during ATP hydrolysis.

It was previously shown that V-ATPase activity, but not V-ATPase mediated acidification of lysosomes, is necessary for mTOR signaling (Zoncu et al, 2011). Furthermore, the C-terminal $\alpha$ helix of mEAK-7 was previously proposed to interact directly with mTOR (Nguyen et al, 2018), which could not occur when mEAK-7 is attached to V-ATPase. The observed ATP-sensitive interaction of mEAK-7 with V-ATPase could relate to its proposed roles: partial displacement of mEAK-7 from V-ATPase during ATP hydrolysis by V-ATPase would allow one of the domains from mEAK-7 to bind other proteins (Fig 4B and Video 1). To detect mEAK-7 interactions that could occur when the protein is fully or partially separated from V-ATPase, we prepared a construct with N-terminally 3× FLAG tagged mEAK-7 and incubated it with both porcine kidney and brain lysate in the presence of 1% CHAPS. We then purified the mEAK-7 by affinity chromatography and identified enriched proteins by mass spectrometry (Fig S8). As a control experiment, proteins from an identical purification in the absence of 3× FLAG-tagged mEAK-7 were also identified. This experiment showed specific enrichment of numerous V-ATPase subunits, but no additional interacting proteins could be detected with confidence. It is possible that interaction of the C-terminal $\alpha$ helix from mEAK-7 with other proteins relies on post-translational modifications, as this region of the protein was found to be phosphorylated at Ser440, 444, and 446 in multiple cellular contexts (Wilhelm et al, 2014). Phosphorylation of these residues would likely prevent the C-terminal domain from binding V-ATPase and could also serve as a mechanism for regulating mEAK-7 binding to V-ATPase and other proteins. Alternatively, separation of mEAK-7 from V-ATPase could expose new binding sites on V-ATPase for other interacting protein. Finally, it is possible that protein interactions with mEAK-7 were missed because of their occurring at sub-stoichiometric levels, with low affinity, or in different cellular contexts.

## Discussion

TLDc domain-containing proteins are widespread in biology (Fig 5A) (Finelli & Oliver, 2017). OXR1 (Merkulova et al, 2015) and NCOA7

(Castroflorio et al, 2021) are known to interact with V-ATPase, with mutation of the latter causing neurological effects. The short TLDc-containing NCOA7-B isoform is induced by interferons and also interacts with V-ATPase, inhibiting endosome-mediated viral entry into cells and apparently increasing vesicle acidification (Doyle et al, 2018).

Following release of a preprint for the present study (Tan et al, 2021 *Preprint*), an article was published showing that the yeast TLDc protein Oxr1p binds to the yeast $V_1$ complex, releasing subunit H and promoting dissociation of $V_1$ from $V_O$ (Khan et al, 2022). Subunit H normally inhibits ATP hydrolysis in the $V_1$ complex (Parra et al, 2000), but $V_1$:Oxr1p is inhibited even though subunit H is absent (Khan et al, 2022). Comparison of the binding mode of Oxr1p (Fig 5B, *left*) and mEAK-7 (Fig 5B, *middle*) shows notable differences in their interactions with V-ATPase: Oxr1p binds the enzyme in rotational State 1, whereas mEAK-7 binds it in rotational State 2. Furthermore, Oxr1p attaches to peripheral stalk 2, whereas mEAK-7 attaches to peripheral stalk 3. Most notably, Oxr1p binding forces the peripheral stalk 2 into a highly strained conformation (Fig 5B, *left*) that is not seen in any of the peripheral stalks in V-ATPase with mEAK-7 (Fig 5B, *center*). This strained peripheral stalk explains why Oxr1p binding promotes dissociation of $V_1$ from $V_O$, whereas mEAK-7 binding does not. Recent structures of yeast $V_1$ and $V_1\Delta C$ complexes show a sharp bending of peripheral stalk 2 induced by interaction with subunit H (Vasanthakumar et al, 2022), which is similar to the bending seen with Oxr1p (Fig 5B, *right*). The interaction of subunit H with peripheral stalk 2 and the resulting bending of the peripheral stalk was proposed to be responsible for inhibiting ATP hydrolysis of $V_1$, as disruption of the interaction prevents subunit H from inhibiting ATP hydrolysis (Vasanthakumar et al, 2022). Thus, the different location for mEAK-7 binding compared with Oxr1p and its minor influence on the structure of the V-ATPase explain why Oxr1p inhibits ATP hydrolysis but mEAK-7 does not.

How the labile binding of mEAK-7 and inhibitory binding of Oxr1p relate to the in vivo activities of the proteins remains unclear and warrants further study. Whereas Oxr1p appears to have evolved to disrupt V-ATPase and block its activity, mEAK-7 could function as a sensor of V-ATPase activity. This variability between the binding mode and effect of TLDc proteins points to interesting differences throughout the TLDc protein family, which have been linked to numerous different processes (Finelli et al, 2019; Svistunova et al, 2019; Castroflorio et al, 2021). Although it has become clear that TLDc domains are V-ATPase interaction modules (Eaton et al, 2021a), the

differences between the proteins suggests a fascinating diversity of biological roles that remains to be uncovered.

## Materials and Methods

### Protein purification

Porcine kidneys were chilled to 4°C in 1 liter of homogenization buffer (4 mM Hepes, pH 7.4, 320 mM sucrose, and 0.2 mM PMSF). All subsequent steps were performed at 4°C. Each kidney was dissected to separate the softer cortex and medulla tissue from the more rigid pelvis tissue. The cortex and medulla tissue were blended with homogenization buffer for 1 min in a total volume of ~500 ml for four kidneys at high speed followed by a 10-s pause and an additional 1 min of blending. Cell debris was removed by centrifugation at 800$g$ for 20 min. The membrane fraction from cells was then collected by centrifugation at 25,000$g$ for 20 min and flash frozen and stored at −80°C. To solubilize the V-ATPase, solubilization buffer (50 mM Hepes, pH 7, 320 mM sucrose, 300 mM NaCl, 10% [vol/vol] glycerol, 5 mM EDTA, 5 mM aminocaproic acid, 5 mM para aminobenzamidine, 0.2 mM PMSF) was added to the membrane fraction at 8 ml/g and resuspended with a Dounce homogenizer. Approximately 6 g of membranes were used for each preparation. The detergent glycol diosgenin (GDN) was added to 1% (wt/vol) and incubated with gentle mixing overnight. After solubilization, V-ATPase purification and assay was performed with no added lipids at 37°C as described previously (Abbas et al, 2020), except with 0.025% (wt/vol) GDN as the detergent and gel filtration with a Superose 6 Increase 10/300 column (GE Healthcare) after affinity purification. Enzyme-coupled ATPase activity assays were performed as described previously (Kornberg & Pricer, 1951; Vasanthakumar et al, 2019). Briefly, enzyme preparations were assayed in a 96-well plate with a total reaction volume of 160 $\mu$l. Purified V-ATPase was added to the ATPase assay reaction buffer (50 mM Tris, pH 7.4, 3 mM MgCl, 0.2 mM NADH disodium salt, 3.2 units pyruvate kinase, 8 units L-lactic dehydrogenase, and 0.025% [wt/vol] GDN) and the reaction was initiated with the addition of 1 mM ATP disodium salt and 1 mM phosphoenol pyruvic acid monopotassium salt. Either bafilomycin A1 in DMSO at 1 $\mu$M or DMSO without bafilomycin was added to each well. Absorbance at 340 nm was monitored at 30°C to measure the signal from NADH and converted to concentration of NADH with a standard curve.

Recombinant mEAK-7 from *Sus scrofa* (UniProt ID: A0A4X1T484) was synthesized in a pET28a(+) vector with an N-terminal 6× His tag (GenScript). The codon optimized sequence for mEAK-7 was:

GGGAATTCAAAAAGTAGGTCTGGACAAGGTCTTTGCAGCCGTTTCCTGCC GGAGGAACAAGCGGAAGTGGACGGTCTGTTCGATGCACTCAGCTCTGAGAA GCTGTCCTCACGGACCAGCCCGCGTAGCTTTAGCCTGCAAGCGCTGAAATC TCACGTAGGCGAAGCGTTGCCACCGGAGATGGTTACGCGTCTGTTCGAGGGT ATGCGTCGTGCGGATCCGACAGGCAAGGCGACCGGTCCGAGCGCACGCATC AGCCAGGAGCAATTTACCTTGAGTATGAGCCATTTGCTGCGCGGCAGCAGC GAGGAAAAATCCTTGGTTATTCTGGCAATGGCTGCCGCTACCGATGGCCCA GCGGAAGCCCGTGAGGTTCTGCGCTTCACGGAAGACCTGGTGGGCAGCGTC GTGCATGTCTTACACTACCGCCAAGAGCTGCGCGGTTGGACCCAGAAACA GGCCTCTGGTTCCCCGCCTCGTGTTCAGGCGTTGGCGGCACAATTATTCTC CGAGCTGAAGCTGCAGGACGGCGAGAAGCTGCCGGGTCCGCAGCGTCTG GACTGCGATTGTGATCGTGCAGTCGTGGAAGCGTGGCTGTTCCGCGCTCCG CATGTTGCAACCTTTCTGTCCGTGGTGATTCATCAGGGGTTTCGCTTGCTGC GCTCCAGCCTGGACTTGGCGACTCTGCTGCCAGAACGTCAAGTTGACCGAG GCCGTGAATTTGCGTCGCTGCTGGACGTGCTGAGCGTTGCCTATATCAACAG TCACCTGCCGCGTGATCTCCGTCATCGTTGGCGTCTCCTTTTCGCCACGGCT CTGCACGGTCACAGCTTTGCTCAATTGTGCGGTCGTATCACCCAGCGCGGC CCGTGCGTGGTGCTGTTGGAAGATCAGGATGGCCACGTTTTTGGCGGCTTCG CGTCTTGTAGCTGGGAAGTGAAACCGCAGTTTCAGGGTGACAGCAAGTGCTT TCTTTTCTCGATCTGCCCGGCTATGGCGGTTTACACCTGCACCGGGTATAACG ATCATTACATGTATCTGAATCATGGTCAGCAGACCATTCCGAATGGTCTGGGT ATGGGTGGTCAACACAACTACTTCGGTCTGTGGGTTGACGTTGATTTTGGTA AAGGTCACTCCAAAGCAAAACCGACCTGTACCACGTACAGCAGCCCACAAC TGTCGGCTCAAGAGGACTTCCGCTTCGAAAAGATGGAAGTTTGGGCAGTGG GCGACCCGTCTGTCACTCAACCGGCGAAAAGCAGCAAGTCCATCCTGGACG GCGACCCGGAGGCGCAAATTCTGCTTGAGGCGAGCGGCAGAAGCCGTCACTC CGAAGGTTTGCGTGCGGTGCCGGAGGACGATTAA.

ArcticExpress competent cells (Agilent) were transformed with the plasmid and plated onto Luria broth (LB) agar supplemented with 50 $\mu$g ml$^{-1}$ kanamycin (Sigma-Aldrich) and grown overnight at 37°C. A transformed colony was used to inoculate a starter culture containing 10 ml of 2xYT medium supplemented with 50 $\mu$g ml$^{-1}$ kanamycin. The starter culture was grown overnight at 37°C with shaking at 240 rpm, and 10 ml of this culture was used to inoculate 1 liter of 2xTY medium supplemented with 50 $\mu$g ml$^{-1}$ kanamycin. The culture was grown at 37°C with shaking at 240 rpm to an OD of ~0.8, with the temperature then reduced to 7°C and protein expression was induced overnight by addition of 1 mM isopropyl $\beta$-D-1-thiogalactopyranoside (IPTG) (Fisher). All further steps were performed at 4°C. Cells were harvested by centrifugation at 4,000$g$ for 15 min and the cell pellet was resuspended in PBS and centrifuged again at 4,000$g$ for 30 min. The cell pellet was then resuspended in lysis buffer (50 mM Tris−HCl, pH 7.4, 300 mM NaCl, and 0.2 mM PMSF) at ~8 ml/g of wet cell pellet. The resuspended cells were lysed by sonication (Q Sonica Q500) with 2 s on, 2 s off at 30% amplitude for 10 min, and cell debris was removed by centrifugation at 130,000$g$ for 45 min with a Type 70 Ti Rotor (Beckman Coulter). The supernatant was filtered through a 0.22 $\mu$m filter (Sigma-Aldrich) and loaded onto a HisTrap HP 5 ml column previously equilibrated with HisTrap buffer (50 mM Tris−HCl, pH 7.4, 300 mM NaCl, and 20 mM imidazole). The column was washed with 10 column volumes (CV) of HisTrap buffer followed by elution with five CV of HisTrap buffer with imidazole added to 300 mM. Eluted protein was concentrated with a 10-kD MWCO concentrator (Amicon) to ~500 $\mu$l before two rounds of gel filtration, first with a Superose 6 Increase 10/300 column and then a Superdex 75 10/300 column. The purified protein was pooled, flash frozen, and stored at −80°C for subsequent use.

### Cryo-EM sample preparation, data collection, and image analysis

Purified V-ATPase was concentrated to ~20 mg/ml and 1.2 $\mu$l was applied onto the copper side of nanofabricated gold grids (Marr et al, 2014) in the environmental chamber of a Leica EM GP2 grid freezing device (>80% RH, 277 K). Grids were blotted on the copper side for 1 s before plunge freezing in liquid ethane. To freeze grids with V-ATPase and mEAK-7, V-ATPase at 20 mg/ml was mixed with a ~20× molar excess of mEAK-7 (~1 mg/ml) and incubated for 30 min

on ice. For grids with V-ATPase, mEAK-7, and ATP, V-ATPase was first incubated with mEAK-7 on ice. ATP with MgCl$_2$ at 50 mM (0.4 $\mu$l) was applied on the grid, and immediately before plunge freezing, 1.6 $\mu$l of V-ATPase:mEAK-7 mixture was added and mixed by pipetting for 2 s. This procedure results in a final concentration of 10 mM ATP and requires a total of 5 s from mixing to plunge freezing.

For the V-ATPase alone, cryoEM data were acquired at 300 kV with a Thermo Fisher Scientific Titan Krios G3 electron microscope and prototype Falcon 4 camera operating in electron counting mode. The calibrated pixel size was 1.02 Å/pixel and 5,455 movies, consisting of 29 exposure fractions each with a total exposure of 40 e$^-$/ Å$^2$, were collected. For V-ATPase with recombinant mEAK-7, mEAK-7ΔCterm, mEAK-7 plus ATP, and mEAK-7 plus EDTA with EGTA, cryoEM data were acquired with the same electron microscope. However, for these datasets, a lower magnification was used with a calibrated pixel size of 1.3225 Å/pixel, and 5,142, 2,013, 5,498, and 2,468 movies consisting of 29 exposure fractions each with total exposures of 36, 37, 44, and 41 e$^-$/Å$^2$, respectively, were collected. For the V-ATPase with mEAK-7 and calcium, a Thermo Fisher Scientific Glacios electron microscope and Falcon 4 camera with Selectris X energy filter was used. The slit width was 10 eV, and aberration-free image shift was used during data collection. The calibrated pixel size was 0.895 Å/pixel and 3,209 movies with 1,288 frames in EER mode (Guo et al, 2020) with a total exposure of 40 e$^-$/Å$^2$ were collected. All data were collected using the EPU data collection software and, except for the sample with calcium added, monitored using *cryoSPARC* Live (Punjani et al, 2017).

For V-ATPase alone, movies were aligned with UCSF MotionCor2 (Zheng et al, 2017) through *Relion* v3 (Scheres, 2012) and processed using *cryoSPARC* v3. CTF parameters were estimated in patches and template selection of particle images performed using the map of rat brain V-ATPase as the reference (Abbas et al, 2020). 2D classification was then performed to remove undesirable particle images. Multiple rounds of ab initio refinement were used to further clean the dataset and a consensus 3D map at 3.9 Å resolution from 133,337 particle images was calculated with non-uniform refinement (Punjani et al, 2020). This consensus set of particle images, with their corresponding orientation parameters, were transferred to *Relion* v3 for particle polishing (Zivanov et al, 2019) and transferred back into *cryoSPARC* for heterogenous refinement to separate the three rotational states and for CTF refinement. Non-uniform refinement was then carried out for each of the three rotational states. For the other datasets, a similar processing strategy was used. For EER data, movies were converted to TIF files containing 46 exposure fractions using *relion_convert_to_tiff*. Transfer of data between *Relion* and *cryoSPARC* was done with *pyem* (10.5281/ zenodo.3576630).

## Exhaustive focused classification

Exhaustive focused classification was performed by adapting the *Relion* signal subtraction and focused classification strategy (Bai et al, 2015). The dataset of particle images, including datasets with additional nucleotide, were classified into the three main rotational states of the enzyme. Next, a model of the enzyme was fitted into the map and used to create a mask for subtracting signal from the core V-ATPase complex from individual particles. A mask that surrounds the V-ATPase complex but extends ~20 Å from the surface of complex was then created to search for weakly bound proteins. This search space was divided into four main regions: the V$_1$ region, the region above V$_1$, the region adjacent to subunit a, and the region surrounding ATP6AP1/Ac45. Focused classification was then performed using each of these four masks. Particle orientation parameters were not refined and a range of T values were tested, as outlined previously (Bai et al, 2015). 15 classes were used during classification. If a potential additional density was identified, a new and tighter mask around the region of interest was created to select the signal of interest.

## Atomic model building

The atomic models from rat brain V-ATPase (Abbas et al, 2020), the H subunit from human V-ATPase (Wang et al, 2020a), and an mEAK-7 model predicted by Phyre2 (Kelley et al, 2015) were used to begin model building. Models were built manually in *Coot* (Emsley & Cowtan, 2004) and refined with iterative rounds of adjustment in *Coot*, *Rosetta relax* (Wang et al, 2016), and real space refinement with *Phenix* (Adams et al, 2010; Afonine et al, 2018). This process was continued until statistics calculated with *Molprobity* (Chen et al, 2010) ceased to improve. The final map and model were validated with *cryoSPARC*'s implementation of *blocres* (Cardone et al, 2013) to assess map local resolution, the *3DFSC* suite (Tan et al, 2017) to calculate directional resolution anisotropy, and the *SCF* program (Baldwin & Lyumkis, 2020) to calculate the sampling compensation factor, which quantifies how preferred particle orientations contribute to attenuation of the Fourier Shell Correlation (FSC). Map-to-model FSCs were calculated by converting the model to a map with the *molmap* function in *Chimera* at the Nyquist resolution. A mask was made from this map with *Relion* (after low-pass filtering to 8 Å, extending by one pixel, and applying a cosine-edge of three pixels) and was applied to the map. Map-to-model FSC was calculated with *proc3d* in *EMAN* (Ludtke et al, 1999).

## Mass spectrometry

Mass spectrometry analysis of the purified V-ATPase and isoform identification was performed as described previously (Abbas et al, 2020). To identify proteins that bind mEAK-7 by mass spectrometry, purified mEAK-7 with 3× FLAG tag was immobilized on M2 affinity gel matrix (2 ml) and washed with 10 CVs of buffer (50 mM Hepes, pH 7, 300 mM NaCl, 10% [vol/vol] glycerol) to remove unbound mEAK-7. Membrane fractions (prepared in the same manner for protein purification) were incubated with 1% (wt/vol) CHAPS for 2 h and passed through the column, followed by 10 CVs of wash buffer (300 mM NaCl, 50 mM Hepes, pH 7, 10% [vol/vol] glycerol, 0.3% [wt/vol] CHAPS), and finally three CVs of wash buffer with 150 $\mu$g/ml 3× FLAG peptide to elute the mEAK-7. The eluted protein was concentrated and prepared for mass spectrometry using S-Trap Micro spin columns (Protifi) according to manufacturer's instructions. Each sample (5 $\mu$l in 5% formic acid) was loaded at 800 nl/min onto an equilibrated HPLC column and peptides were eluted over a 90-min gradient generated by a Eksigent ekspertTM nanoLC 425 (Eksigent) nano-pump and analysed on a TripleTOF 6600 instrument (SCIEX)

operated in data-dependent acquisition (DDA) mode. The DDA method consisted of one 250 ms MS1 TOF survey scan from 400 to 1,800 D followed by 10 100 ms MS2 candidate ion scans from 100 to 1,800 D in high sensitivity mode. Only ions with a charge of 2+ to 5+ that exceeded a threshold of 300 cps were selected for MS2, and former precursors were excluded for 7 s after one occurrence. To minimize carryover between each sample, the analytical column was washed for 2 h by running an alternating sawtooth gradient from 35% acetonitrile with 0.1% formic acid to 80% acetonitrile with 0.1% formic acid at a flow rate of 1,500 nl/min, holding each gradient concentration for 5 min. Analytical column and instrument performance were verified after each sample by loading 30 fmol BSA tryptic peptide standard with 60 fmol α-casein tryptic digest and running a short 30 min gradient. TOF MS mass calibration was performed on BSA reference ions before running the next sample to adjust for mass drift and verify peak intensity.

All raw (WIFF and WIFF.SCAN) files were saved in ProHits (Liu et al, 2016). mzXML files were generated from raw files using the ProteoWizard converter converter (v3.0.4468) and SCIEX converter (v1.3 beta), implemented within ProHits. The searched databases contained the pig (v88) or rat (v97) complement of the RefSeq protein database (including reversed sequences and common contaminants: 128,008 and 134,277 total sequences, respectively). mzXML files were searched by Mascot (v2.3.02) and Comet (v2016.01 rev. 2) using the following parameters: up to two missed trypsin cleavage sites, methionine oxidation and asparagine/glutamine deamidation as variable modifications. The fragment mass tolerance was 0.15 D and the mass window for the precursor was ±40 ppm with charges of 2+ to 4+ (both monoisotopic mass). Search engine results were analysed using the Trans-Proteomic Pipeline (TPP v4.6 OCCUPY rev 3) via iProphet. Spectral count data were visualized using ProHits-viz (https://prohits-viz.org/) (Knight et al, 2017).

### Localization of mEAK-7 and ratiometric determinations of lysosomal and phagosomal pH

The sequence for mEAK-7 was amplified from the pET28a(+) vector from GenScript for Gibson assembly (New England Biolabs), cloning into the pmCherry-N1 (Clonetech) expression vector for mammalian cells with the forward primer GTCGCCACCATGgggaattcaaaaagtaggtctggacaaggtctttgcagccgtttc and the reverse primer CTCGCCCTTGCTCACatcgtcctccggcaccgcacgcaaaccttcggagtgacggcttct, which maintains the C-terminal mCherry tag. The reverse primer CTCGCCCTTGCTCACttaatcgtcctccggcaccgcacgcaaaccttcggagtgacggct was used to create a plasmid with a stop codon before the mCherry tag for expression of mEAK-7 without the tag. HeLa and HEK293T cells were seeded on 18 mm glass coverslips at a density of $3 \times 10^5$ cells/ml and were transfected with FuGENE 6 (Promega) transfection reagent 24 h later. RAW264.7 cells were seeded on 18 mm glass coverslips at a density of $5 \times 10^5$ cells/ml and transfected 24 h later with FuGENE HD. Cells were transfected with 3 $\mu$l of FuGENE per 1 $\mu$g of total plasmid. Both cell types were imaged 18–24 h post-transfection.

To visualize lysosomal compartments, HeLa cells transiently expressing mEAK-7-mCherry were incubated with 250 $\mu$g/ml fluorescein isothiocyanate (FITC)-conjugated 10 kD-dextran at the time of transfection and chased with complete medium 1 h before imaging. HEK293T cells were transiently transfected with mEAK7-mCherry and LAMP1-GFP to label the lysosomes. Coverslips with cells were mounted in a Chamlide magnetic chamber, incubated at 37°C in HBSS medium, and visualized with a Zeiss Axiovert 200 M confocal microscope operated by Volocity v6.3 software. Images were acquired using a 63×/1.4 NA oil objective (Zeiss) with an additional 1.5× magnifying lens and processed using Volocity v6.3 and Adobe Illustrator.

To measure pH of lysosomes, HEK293T cells transiently transfected with unlabelled mEAK-7 and PLCδ-PH-RFP (to allow identification of transfected cells) were incubated with 250 $\mu$g/ml FITC-conjugated 10-kD dextran at the time of transfection and chased with complete medium 1 h before imaging to visualize lysosomal compartments. RAW264.7 cells co-transfected with unlabelled mEAK-7 and PLCδ-PH-RFP were incubated with FITC-conjugated zymosan particles, which were taken up into phagosomes, 1 h before imaging and were chased with complete medium 30 min before imaging to wash away unbound particles. Steady-state lysosomal and phagosomal pH were determined by exciting FITC labelled lysosome/phagosomes at 481 ± 15 nm and 436 ± 20 nm, respectively, and collecting emitted light at 520 ± 35 nm. The pH-dependent fluorescence intensity of FITC when excited at ~490 nm was used as described below to determine the steady-state pH of lysosomes. The relatively pH-insensitive fluorescence intensity of FITC when excited at ~440 nm was used to control for photobleaching during image acquisition. Multiple fields of cells were imaged for each condition, and the data were processed with Volocity v6.3. To convert fluorescence ratios to pH values, cells were sequentially subjected to isotonic K$^+$ solutions (143 mM KCl, 5 mM glucose, 1 mM MgCl$_2$, 1 mM CaCl$_2$) at different pH values (pH 4.5 solution buffered with 20 mM acetic acid, pH 5.5, 6.5, and 7.5 solutions buffered with 20 mM MES), containing 10 $\mu$M nigericin and 5 $\mu$M monensin. Cells were imaged 5 min after the addition of each solution to obtain the 490 nm/440 nm fluorescence ratio corresponding to each pH standard. A calibration curve was prepared from a least-squares fit of mean background-subtracted fluorescence ratios as a function of pH. The pH measurements were compared with an unpaired two-tailed $t$ test.

## Data Availability

CryoEM maps have been deposited into the EMDB with accession codes EMD-26385, EMD-26386, EMD-26387, and EMD-26388. Models have been deposited into the PDB with accession codes 7U8O, 7U8P, 7U8Q, and 7U8R. Mass spectrometry data for proteins that bind mEAK-7 were deposited into the ProteomeXchange with accession code PXD034953, through partner MassIVE with accession code MSV000089750.

## Supplementary Information

# Life Science Alliance

# Acknowledgements

We thank S Benlekbir for assistance with cryo-EM data collection, Z-Y Lin for assistance with mass spectrometry sample preparation and analysis, G Wasney and S Popa assistance with circular dichroism, A Kotecha and FJA Koh for data collection for the V-ATPase sample with mEAK-7 and calcium using the Glacios microscope with Falcon4 Selectris X, B McDougall for providing porcine tissue, and members of the Rubinstein laboratory for discussions. This work was supported by the Canadian Institutes of Health Research (CIHR) Project Grant PJT166152 (JL Rubinstein), CIHR Foundation grants FDN-143202 (S Grinstein), FDN-143301 (A-C Gingras), the Canada Research Chairs program (JL Rubinstein and A-C Gingras), Canadian Cancer Society Research Institute i2I grant 705938 (A-C Gingras), Wellcome Trust, grant 221795/Z/20/Z (CV Robinson), Canadian Institutes of Health Research Postdoctoral fellowship (YZ Tan, YM Abbas), Agency for Science, Technology and Research Singapore (YZ Tan), National University of Singapore (NUS) Presidential Young Professorship R-154-000-C62-133 (YZ Tan), and Ministry of Education Singapore (YZ Tan). Cryo-EM data were collected at the Toronto High-Resolution High-Throughput cryo-EM facility and enzyme assays performed using infrastructure from the Hospital for Sick Children's Structural & Biophysical Core Facility, both of which are supported by the Canada Foundation for Innovation and Ontario Research Fund. Some of the proteomics experiments were performed at the Network Biology Collaborative Centre at the Lunenfeld-Tanenbaum Research Institute, a facility supported by Canada Foundation for Innovation funding, by the Government of Ontario, and by Genome Canada and Ontario Genomics (OGI-139).

## Author Contributions

YZ Tan: conceptualization, data curation, formal analysis, investigation, methodology, and writing—original draft.
YM Abbas: methodology.
JZ Wu: formal analysis and investigation.
D Wu: formal analysis and investigation.
KA Keon: formal analysis and validation.
GG Hesketh: formal analysis and investigation.
SA Bueler: investigation and methodology.
A-C Gingras: resources and supervision.
CV Robinson: resources and supervision.
S Grinstein: resources and supervision.
JL Rubinstein: conceptualization, resources, formal analysis, supervision, investigation, visualization, methodology, project administration, and writing—original draft, review, and editing.

## Conflict of Interest Statement

The authors declare that they have no conflict of interest.

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
