## [Reviewer comments · Life Science Alliance]

Life Science Alliance

CryoEM of endogenous mammalian V-ATPase interacting with the TLDC protein mEAK-7

Yong Zi Tan, Yazan Abbas, Jing Ze Wu, Di Wu, Kristine Keon, Geoffry Hesketh, Stephanie Bueler, Anne-Claude Gingras, Carol Robinson, Sergio Grinstein, and John Rubinstein

DOI: 10.26508/lsa.202201527

Corresponding author(s): John Rubinstein, The Hospital for Sick Children

Review Timeline:

Submission Date:	2022-05-18
Editorial Decision:	2022-05-19
Revision Received:	2022-05-19
Editorial Decision:	2022-06-13
Revision Received:	2022-06-17
Accepted:	2022-06-21

Transaction Report:

Please note that the manuscript was reviewed at Review Commons and these reports were taken into account in the decision-making process at Life Science Alliance.

May 19, 2022

Re: Life Science Alliance manuscript #LSA-2022-01527

John L Rubinstein
The Hospital for Sick Children
Molecular Structure and Function Program
555 University Avenue, Rm. 3330
Toronto, ON M5G 1X8
Canada

Dear Dr. Rubinstein,

Thank you for submitting your manuscript entitled "CryoEM of endogenous mammalian V-ATPase interacting with the TLDc protein mEAK-7" to Life Science Alliance. We invite you to re-submit the manuscript, revised according to your Revision Plan.

Thank you for this interesting contribution to Life Science Alliance. We are looking forward to receiving your revised manuscript.

Sincerely,

B. MANUSCRIPT ORGANIZATION AND FORMATTING:

Full Revision

1. General Statements [optional]

This section is optional. Insert here any general statements you wish to make about the goal of the study or about the reviews.

This section is mandatory. Please insert a point-by-point reply describing the revisions that were already carried out and included in the transferred manuscript.

Reviewer comments are in Blue. Our responses are in Black.

Reviewer #1 (Evidence, reproducibility and clarity (Required)):

This study brings together cryo-EM, Mass spectrometry and some well considered molecular biology to discuss a new binding partner of the V-ATPase (mEAK-7) and map out its binding interactions. The combination of techniques provides a powerful tool to convince the reader of the existence of the binding and provides some interesting questions for future studies in the field.

Major comments: The manuscript is very well written and data well presented I do not have any major concerns with the study.

Minor comments: There are a few minor comments which may aid in the clarity of the manuscript.

We thank the reviewer for their enthusiasm, and for pointing out the powerful combination of techniques that we applied.

The introduction is currently very brief consisting of 3 paragraphs, one of which discusses the work reported and not background to the area. Expansion of this may help broaden the scope and allow those not directly in the field to have a better understanding of the work.

We have expanded the introductory text of the manuscript to make it easier to understand by those not in the field: Specifically, we have added a paragraph of background about V-ATPases (p. 1, 2nd paragraph):

“V-ATPase-driven acidification is necessary for targeting and post-translational modification of proteins in the Golgi (Kellokumpu, 2019), degradation of material in lysosomes (Mindell, 2012), and the uptake of cargo for secretory vesicles including synaptic vesicles (Hnasko and Edwards, 2012). V-ATPases targeted to the plasma membrane of specialized cells are responsible for acidifying the extracellular environment (Breton and Brown, 2013; Qin et al., 2012; Stransky et al., 2016). Because of their central role in acidification of intracellular compartments in all cells and acidification of the extracellular environment in specialized cells, complete disruption of V-ATPase activity is embryonic lethal in mammals (Sun-Wada et al., 2000) while aberrant activity or expression is associated with several diseases. For example, defects in V-ATPase-mediated lysosomal degradation is linked to neurodegenerative diseases (Colacurcio and Nixon, 2016) with tissue-specific mutations associated with osteopetrosis (Kornak et al., 2000), and distal renal tubular acidosis (Karet et al., 1999; Smith et al., 2000).”

And a paragraph about TLDC proteins (p. 2, 2nd paragraph):

“In yeast, the small soluble protein Oxr1p consists primarily of a TLDC domain alone and was recently found to promote disassembly of the yeast V-ATPase V₁ and V₀ regions (Khan et al., 2021). In contrast, numerous mammalian TLDC domain-containing proteins also possess additional domains (Finelli and Oliver, 2017). These include a polysaccharide-binding Lysine Motif (LysM) and protein- or lipid-binding GRAM domain in NCOA7 and OXR1, a TBC domain similar to those found in Rab-GTPase activating proteins in TBC1D24, and a myristoylation motif and apparent calcium-binding EF-hand motif in mEAK-7. The biological functions of these mammalian TLDC proteins remain obscure, but mutations of the genes encoding them can cause diseases, such as a range of neurological disorders upon mutation of the gene for TBC1D24 (Finelli et al., 2019).”

The authors mention 91% inhibition by bafilomycin but I could not see at what concentration this was added.

We apologize for this omission. We now provide the bafilomycin concentration:

“The preparation is also highly coupled, showing 91% inhibition following addition of 1 μ M bafilomycin.”

pg5 line 4, it is stated that mEAK-7 is bound in state 2 but what % have it bound. this would be useful as it could then be contrasted with the low level in the native tissue.

We now state the fraction bound by adding the following text (p. 5, 3rd paragraph):

“The map of rotational State 2 showed V-ATPase with mEAK-7 bound (Fig. 2A, *left*, Fig. S6A and B, Table S2 and S3). Of the 86,338 particle images in the dataset, 43,251 (23%) contributed to maps of V-ATPase in state 2, and all of these images (ie. 23% of total particle images) had density for mEAK-7. This statistic is in sharp contrast to the occupancy of mEAK-7 from native source, where 24.8% of particle images were found in State 2 but only 6.4% of them (1.6% of total particle images) had density for mEAK-7.”

When describing the binding of mEAK-7 are any of these residues unique to certain subunit isoforms? may this show how different isoforms may respond to different cellular stimuli?

Based on the reviewer’s suggestion, we looked at the sequence alignment for subunit isoforms B1 and B2, as the other binding sites of mEAK-7 (subunit D and A) do not have multiple

Full Revision

isoforms. As shown below, residues involved in binding mEAK-7 did not vary between isoforms. We therefore add the sentence to the manuscript:

“The residues in B1 and E1 identified as interacting with mEAK-7 are conserved in the B2 and E2 subunit isoforms, suggesting that mEAK-7 binding is isoform insensitive.”

CLUSTAL O(1.2.4) multiple sequence alignment

```
B1 MAMKVDSPGGLPSSGSHLGTAREHVQVTRNYITHPRVYRTVCSVNGPLVLDHVKFA 60
B2 MATQVDRPSGFTSNCDPGTAREHVQVTRNYITHPRVYRTVCSVNGPLVLDQVKFA 60
   ** :*. **.* : *... *****;*****;*****;*****;*****;*****
B1 QYAEIVNFTLPNGTQRSQVLEVSGTKAIVQVFEGTSGIDAQKTTCEFTGDLIRTPVSED 120
B2 QYAEIVNFTLPDGTQRSQVLEVAGTKAIVQVFEGTSGIDSQKTTCEFTGDLIRTPVSED 120
   *****;*****;*****;*****;*****;*****
B1 MLGRVFNKSGKPIDNGPVVMAEDFLDINGQPINPHDRIVEPEMIQTGISPIDVMNSIARG 180
B2 MLGRIFNKGSKPIDKGPVAVMAEDFLDINGQPINPHDRIVEPEMIQTGISPIDVMNSIARG 180
   ***;*****;*. *****;*****;*****;*****;*****
B1 QKIPFSAAGLPHNEIAAQICRQAGLVKSKAVLDYDDNFVIFAAMGVNMETARFFKS 240
B2 QKIPFSAAGLPHNEIAAQICRQAGLVKSKAVLDYHEDNFVIFAAMGVNMETARFFKS 240
   *****;*****;*****;*****;*****;*****
B1 DFEENGTMGNVCLFNLANDPTIERIITPRALTTAEFLAYQCEKHLVILTDMSYAEA 300
B2 DFEQNGTMGNVCLFNLANDPTIERIITPRALTTAEFLAYQCEKHLVILTDMSYAEA 300
   ***;*****;*****;*****;*****;*****
B1 LREVSAAAREEVPGRGPGYMYTDLATIVERAGRVEGRGGSITQIPILTMPNDITHP 360
B2 LREVSAAAREEVPGRGPGYMYTDLATIVERAGRVEGRGGSITQIPILTMPNDITHP 360
   *****;*****;*****;*****;*****;*****
B1 DLTFGITTEGQIYVDRQLHNRQIYPPINVLPSLSRLMKSALGEGMTRKDHGDSNQLYACY 420
B2 DLTFGITTEGQIYVDRQLHNRQIYPPINVLPSLSRLMKSALGEGMTRKDHGDSNQLYACY 420
   *****;*****;*****;*****;*****;*****
B1 AIGKDYQAMKAVVGEALTSSEDLLEYFLQKFERNFINQGPYKRRVFSFESLDLWKLRLI 480
B2 AIGKDYQAMKAVVGEALTSSEDLLEYFLQKFERNFINQGPYENRTVFSFESLDLWKLRLI 480
   *****;*****;*****;*****;*****;*****
B1 FPKEMLKRIPQSMIDEFYSREGAPQDPEPEPDTAL 515
B2 FPKEMLKRIPQSVTDEFYSRQGAQQDPASDTAL-- 513
   *****;*****;*****;*****;*****;*****
```

CLUSTAL O(1.2.4) multiple sequence alignment

```
E1 MALSDADVQKIKHMMAFIEQANEKAEIIDAKAEFFNIEKGRVLVQTQRLKIMEYYEKK 60
E2 MALSDVDVQKIKHMMAFIEQANEKAEIIDAKAEFFNIEKGRVLVQTQRLKIMEYYEKK 60
   *****;*****;*****;*****;*****;*****
E1 EKQIEQQKKIQMSNLMNQARLKVLRARDLITDLLNEAKQRLGKVVKDTTRYQVLLDGLV 120
E2 EKQIEQQKKIQMSTMRNARLKVLRARDLITSELLNDAKLSRSIVADQEVYQALLDKLV 120
   *****;*****;*****;*****;*****;*****
E1 LQGLYQLLEPRMIVRCRKQDFLVAQVQKAI PVYK IATKRDVDVQIDQEAYLPEEIAAG 180
E2 LQGLRLLEPVVIRCRPQDLFLVKAQVQKAI PQYTTISHKHVEVQVDQEVQLATDAAG 180
   *****;*****;*****;*****;*****;*****
E2 VEIYNGDRKIKVSNLLESRLDLIAQQMPEVVRGALFGANANRKF 226
E2 VEVYSGDQRIKVSNTLESRLDLLFQQKMPERKALFGANANRKYFV 226
   **.*.*****;*****;*****;*****;*****;*****
```

mEAK-7 expression neither inhibits or activates but could there be other proteins involved which are not purified but form a larger, more stable complex? Overexpression of mEAK-7 alone may not be enough to see this effect as other proteins are still at their native level. I do not think the authors should do any additional experiments as this is beyond the scope but could be discussed. Fascinating to think of other larger complexes that may form to control the V-ATPase, maybe the membrane component would be required as well.

Full Revision

The reviewer raises an interesting point. We have added the following text to introduce this possibility, as well as a possibility introduced by Reviewer 3 (p. 7, 2nd paragraph):

“It is possible that porcine mEAK-7 expressed in human cells does not interact with human V-ATPase in precisely the same way as it does with porcine V-ATPase. However, it is worth noting that all the mEAK-7 residues identified above as interacting with V-ATPase subunits are conserved between porcine and human mEAK-7. Further, it is also possible that mEAK-7 perturbation of lysosomal or phagosomal pH requires additional proteins that were not over-expressed here.”

When discussing the full dissociation of mEAK-7 from the V-ATPase in the presence of ATP is one limitation not that it does not dissociate but the 20 fold excess promotes re-association and is therefore still seen bound?

We apologize for not being more clear about the experiments we did to test the removal of mEAK-7 from V-ATPase by ATP, which indeed considered the possibility that the reviewer raises. We have updated the text on to indicate this information (p. 8, 2nd paragraph):

“Despite several biochemical experiments, we were unable to demonstrate that mEAK-7 can be fully dissociated from V-ATPase by the addition of ATP to the complex, including experiments where we washed immobilized V-ATPase:mEAK-7 extensively with a buffer containing ATP before elution of V-ATPase. These experiments suggest that mEAK-7 can remain associated to V-ATPase, either by its TLDC domain or its C-terminal α helix, during ATP hydrolysis.”

Reviewer #1 (Significance (Required)):

The paper provides an interesting new direction to the study of the V-ATPase. Most conventional structural biology studies are reductionist, removing the protein from the native environment which has knock-on effects, especially when studying more loosely associated proteins/regulators. This provides new evidence on regulation and functions, broadening the field. This provides new evidence to the field and reinforces some current studies. The audience will primarily be within the cryo-EM field although the blend of techniques and development of processing algorithms will likely appeal to a broader audience and could be applied to many other systems.

My expertise are within structural biology and the V-ATPase field

We thank the reviewer for their enthusiasm, and for pointing out how our ‘bottom up’ approach not only supports some recent studies, but broadens the field of V-ATPase biology and will appeal to a broader audience of structural biologists.

Reviewer #2 (Evidence, reproducibility and clarity (Required)):

Tan et al. reported the function of endogenous proteins on V-ATPase from porcine kidney using cryo-EM combined with exhaustive focused classification. mEAK-7, a protein involved in dauer formation in nematodes and mTOR signaling showed that the TLDC domain interacted with V-ATPase's stator while its C-terminal alpha-helix bound V-ATPase's rotor, but the protein did not show clear inhibition in rotary catalysis. Unlike the inhibition of yeast V-ATPase by the TLDC

Full Revision

protein Oxr1p, the exogenous mEAK-7 did not inhibit the rotary catalysis, and intracellular overexpression of mEAK-7 proteins did not change the pH of lysosomes or phagosomes. These observations suggest that inhibition by proteins containing the TLDC domain has effects ranging from strong inhibition to weak regulation sensitive to the enzyme's activity.

We agree with the reviewer that our results reveal the diverse effects of TLDC-domain proteins on V-ATPases.

Major comments:

- Are the key conclusions convincing?

The persuasive power is relatively weak because the actual function of mEAK-7 was not revealed.

We agree with the reviewer that the function of mEAK-7, which has been studied extensively by others, is not revealed by our work. However, we do not make any conclusions about the physiological role of mEAK-7 in our manuscript. Therefore, we believe it is a little unfair to say the “persuasive power is relatively weak because the actual function of mEAK-7 was not revealed” when we are not trying to persuade anyone about the function of mEAK-7. In contrast, we believe the conclusions that we *do* make are strongly supported by our data (as the reviewer points out below: “The authors should not qualify their claims to be preliminary or speculative, as they explain the facts based on their results”). In addition to our finding that mEAK-7 does not inhibit V-ATPase, which reveals the diversity of effects of TLDC protein binding on V-ATPases, our manuscript makes other contributions such as development of a method to reveal scarce endogenous interacting proteins by cryoEM and demonstration of that endogenous mEAK-7 interacts with V-ATPase in tissues. We believe that these are important and meaningful contributions.

- Should the authors qualify some of their claims as preliminary or speculative, or remove them altogether?

The authors should not qualify their claims to be preliminary or speculative, as they explain the facts based on their results.

We thank the reviewer for supporting the rigour of our studies.

- Would additional experiments be essential to support the claims of the paper? Request additional experiments only where necessary for the paper as it is, and do not ask authors to open new lines of experimentation.

To strengthen the claims of the paper, it may be necessary to show negative controls such as mEAK-7 knock out cells.

We agree with the reviewer that knockdown of mEAK-7 is an important experiment. However, these experiments were done previously by other. We now emphasize the discussion of those other experiments (p. 3, 3rd paragraph):

Full Revision

“Cells with mEAK-7 knocked down display abnormal mTOR signaling (Nguyen et al., 2018) and an mEAK-7 knockout mouse displays abnormal epididymis, lung, and skin morphology (<https://www.mousephenotype.org/data/genes/MGI:1921597>).”

Further, as discussed in detail below, the HEK293T cells we use in our experiments have negligible expression of mEAK-7, indicating that a knockout experiment would not be informative. Indeed, this lack of endogenous mEAK-7 in mEAK-7 cells is an advantage because it simplifies interpretation of our mEAK-7 expression experiments. As described below, we add new data to show the successful expression of mEAK-7 in this cell background.

- Are the suggested experiments realistic in terms of time and resources? It would help if you could add an estimated cost and time investment for substantial experiments.

As the function of mEAK-7 on V-ATPase, authors showed mEAK-7 overexpression in cells does not alter lysosomal or phagosomal pH. I think it is realistic to carry out the suggested experiment.

We refer to our previous comment that the suggested experiment has been done already (knockdown of mEAK-7)

The previous work showed that mEAK-7 expression in the HEK-293T cells that we used is barely detectable (from Fig. 1 of Nguyen *et al.* 2018, *Science Advances* 4: eaao5838. Red box is ours) and in our own hands we were not able to detect endogenous mEAK-7 at all in HEK-293T cells using the same reagents as Nguyen *et al.* (2018).

To make this point clear we have added the following text (p. 7, 2nd paragraph):

“Endogenous mEAK-7 expression in HEK293T cells is nearly undetectable (Nguyen et al., 2018) so that any mEAK-7 in the cells is expected to be due to over-expression from the vector.”

- Are the data and the methods presented in such a way that they can be reproduced?

The data and the methods were described in detail and presented for reproducibility.

- Are the experiments adequately replicated and statistical analysis adequate?

I think so.

Minor comments:

- Specific experimental issues that are easily addressable.

Full Revision

None

- Are prior studies referenced appropriately?

Yes

- Are the text and figures clear and accurate?

1) Map densities are not clear in Fig. 2E and F.

We have modified the figure as suggested to make the map densities more clear.

2) The movie is 1:34 minutes in total, but the animation ends at 0:50.

We have corrected this problem with the movie file.

- Do you have suggestions that would help the authors improve the presentation of their data and conclusions?

As mentioned above, it may be necessary to show negative controls such as mEAK-7 knock out cells to reinforce the claims of the paper.

As we explain above, endogenous mEAK-7 expression in the HEK293T cells that we use for probing mEAK-7 interaction with V-ATPase is undetectable and as a result knock out or knock down experiments would not add information.

Full Revision

Reviewer #2 (Significance (Required)):

- Describe the nature and significance of the advance (e.g. conceptual, technical, clinical) for the field.

The exhaustive focus classification used in this paper was able to visualize very small amounts of components bound in the main protein complex at higher resolution. This is an important technological advance in this research field.

We thank the reviewer for their enthusiasm, and for pointing out the important technical advance of exhaustive focused classification.

- Place the work in the context of the existing literature (provide references, where appropriate).

This paper already provides sufficient references.

- State what audience might be interested in and influenced by the reported findings.

The extreme method of the exhaustive focus refinement is of technical interest to structural biology researchers.

We again thank the reviewer.

- Define your field of expertise with a few keywords to help the authors contextualize your point of view. Indicate if there are any parts of the paper that you do not have sufficient expertise to evaluate.

Keywords of my field of expertise: cryo-electron microscopy, single particle analysis, structural biology, membrane protein, exhaustive focused classification

Parts that I don't have sufficient expertise: function of the TLDC domain-containing proteins

Reviewer #3 (Evidence, reproducibility and clarity (Required)):

The manuscript. Tan et al., address the mEAK-7 as a new binding partner to V-ATPase and it may participate in the regulation of the proton pump system based on structural and biochemical analyses. The structures of V-ATPase determined with mEAK-7 are novel and further experiments are performed adequately to support the findings, but the functional relationship of their interaction remains still elusive. It would be better to include further description in some points.

We agree (as we did with Reviewer #1) that the function of mEAK-7 is not revealed by our work but reassert that we make other contributions:

- 1) Development of a method that reveals scarce endogenous interacting proteins by cryoEM
- 2) Demonstration that endogenous mEAK-7 interacts with V-ATPase in tissues

3) Demonstration that different TLDC proteins can affect V-ATPase activity in dramatically different ways

As described below, we have added more detail as the reviewer suggests.

Major comments:

1. For Figure 3, The expression level of mEAK-7 in HEK293T cells should also be assessed with western blot or other types of assay, both for WT and mEAK overexpressed.

As described above, the endogenous expression level for mEAK-7 in HEK293T cells is known to be negligible, which dramatically simplifies the interpretation of our experiments. We attempted to use a commercially available antibody against mEAK-7 (the same one used by Nguyen et al., 2018) to probe expression of mEAK-7. Despite exploring numerous conditions we were not able to detect mEAK-7 specifically by western blotting. However, we *know* that mEAK-7 is expressed from the plasmid because we can see the tagged protein by fluorescence microscopy. We realize that we had not previously demonstrated that expression in HEK293T cells results in expression and correct localization of mEAK-7 as the results we showed previously with mCherry tagged mEAK-7 were from HeLa cells. We now include fluorescence microscopy images that show successful expression and correct localization of mEAK-7 in HEK293T cells as part of Figure 3.

It is possible that the antibody cannot detect the mEAK-7 because the antibody was raised against human mEAK-7 while we are expressing porcine mEAK-7. Although this possibility seems unlikely due to the high sequence identity between the two species (shown below in

response to the next point), we now point out that porcine mEAK-7 may bind differently to human mEAK-7 when interacting with the human V-ATPase (p. 7, 2nd paragraph):

“It is possible that porcine mEAK-7 expressed in human cells does not interact with human V-ATPase in precisely the same way as it does with porcine V-ATPase. However, it is worth noting that all the mEAK-7 residues identified above as interacting with V-ATPase subunits are conserved between porcine and human mEAK-7. Further, it is also possible that mEAK-7 perturbation of lysosomal or phagosomal pH requires additional proteins that were not over-expressed here.”

2. Reconstruction of the expression vector of mEAK-7 for HEK cell expression should be described, including the information of the gene and plasmid used. If the mEAK-7 gene used for the cell assay is from *Sus scrofa*, the authors should concern whether the result resembles in vivo environments indeed or could be affected by chimeric nature of V-ATPase and mEAK-7.

We thank the reviewer for pointing out this omission. We now describe the reconstruction of the expression plasmid:

“Recombinant mEAK-7 from *Sus scrofa* (uniprot ID: A0A4X1T484) was synthesized in a pET28a(+) vector with an N-terminal 6xHis tag (GenScript).”

And

“The sequence for mEAK-7 was amplified from the pET28a(+) vector from GenScript for Gibson assembly (New England Biolabs), cloning into the pmCherry-N1 (Clontech) expression vector for mammalian cells with the forward primer GTCGCCACCATGgggaattcaaaaagtaggtctggacaaggctttgcagccgttc and the reverse primer CTCGCCCTTGCTCACatcgtctccggcaccgcacgcaaaccttcggagtgacgcttct, which maintains the C-terminal mCherry tag. The reverse primer CTCGCCCTTGCTCACtaatcgtctccggcaccgcacgcaaaccttcggagtgacggct was used to create a plasmid with a stop codon before the mCherry tag for expression of mEAK-7 without the tag”.

We also now discuss the possibility that porcine mEAK-7 may bind differently to human mEAK-7 when interacting with the human V-ATPase. However, we point out that all the residues identified in mEAK-7 as interacting with V-ATPase are conserved between human and porcine mEAK-7 (p. 7, 2nd paragraph):

“It is possible that porcine mEAK-7 expressed in human cells does not interact with human V-ATPase in precisely the same way as it does with porcine V-ATPase. However, it is worth noting that all the mEAK-7 residues identified above as interacting with V-ATPase subunits are conserved between porcine and human mEAK-7. Further, it is also possible that mEAK-7 perturbation of lysosomal or phagosomal pH requires additional proteins that were not over-expressed here.”

Porcine	1	MGNSKSRSGQGLCSRFLPEEQAEVDGLFDALSSEKLSRSTSPRSFSLQALKSHVGEALPP	60
		MGNS+SR G+ CS+FLPEEQAE+D LFDALSS+K S S +SFSL+AL++HVGEALPP	
Human	1	MGNSRSRVGRSFCSQFLPEEQAEIDQLFDALSSDKNSPNVSSKSFSLKALQNHVGEALPP	60
Porcine	61	EMVTRLFEGMRRADPTGKATGPSARISQEQFTLSMSHLLRGSSEEKSLVILAMAAATDGP	120
		EMVTRL++GMRR D TGKA GPS +SQEQFT SMSHLL+G+SEEKSL+I+ M +AT+GP	
Human	61	EMVTRLYDGMRRVDLTGKAKGPSENVSQEQFTASMSHLLKGNSEEKSLMIMKMSATEGP	120
Porcine	121	AEAREVLRFTEDLVGSVVHVLHYRQELRGWTQKQASGSPPRVQALAAQLFSELKLQDGEK	180
		+AREV +FTEDLVGSVVHVL +RQELRGWT K+A G PRVQ LAAQL SE+KLQDG++	

Human	121	VKAREVQKFTEDLVGSVVHVLSHRQELRGWTGKEAPGPNPRVQVLAAQLLSEMKLQDGKR	180
Porcine	181	LPGPQRLLDCDCDRAVVEAWLFRAPHVATFLSVVIHQGFrlrrssldlatllPERQVDRGR	240
		L GPQ LD DCDRAV+E W+FR PHVA FLSVVI +GF +L SSLDL TL+PERQVD+GR	
Human	181	LLGPQWLDYDCDRAVIEDWVFRVPHVAIFLSVVICKGFLVLCSSLDLTTLVPERQVDQGR	240
Porcine	241	EFASLLDVLSVAYINSHLPRDLRHRWRLLFATALHGHSAQLCGRITQRGPCVVLLEDQD	300
		F S+LDVLSV YIN+ LPR+ RHRWRLLF++ LHGHSF+QLCG IT RGPCV +LED D	
Human	241	GFESILDVLSVMYINAQLPREQRHRWRLLFSSLEHGHSAQLCGHITHRGPCVAVLEDHD	300
Porcine	301	GHVFGGFASCSWEVKPQFQGDSKCFLFSICPAMAVYTCTGYNDHYMYLNHGQQTIPNGLG	360
		HVFGGFASCSWEVKPQFQGD++CFLFSICP+MAVYT TGYNDHYMYLNHGQQTIPNGLG	
Human	301	KHVFGGFASCSWEVKPQFQGDNRCLFSICPSMAVYTHTGNDHYMYLNHGQQTIPNGLG	360
Porcine	361	MGGQHNIFGLWVDVDFGKGHSAKPTCTTYSSPQLSAQEDFRFEKMEVWAVGDPSVTQPA	420
		MGGQHNYFGLWVDVDFGKGHSAKPTCTTY+SPQLSAQE+F+F+KMEVWAVGDPS Q A	
Human	361	MGGQHNIFGLWVDVDFGKGHSAKPTCTTYNSPQLSAQENFQFDKMEVWAVGDPSEEQLA	420
Porcine	421	KSSKSILGDPEAQILLEASGRSRHSEGLRAVPEDD	456
		K +KSILD DPEAQ LLE SG SRHSEGLR VP+D+	
Human	421	KGNSILDADPEAQALLEISGHSRHSEGLREVDPDE	456

3. Though the error bar scale is n.s. in Fig3C and 3D, the tendency (and the pH range overall) of WT and mEAK overexpressed does not seem to be similar to each other. The authors should explain more about the statistical analyses for the data, as well as the experimental processes such as multiplication.

We now describe the analysis in detail, including the statistical test. The differences the Reviewer refers to are simply from inclusion of a different number of points from different cells, which is not something the statistical test is sensitive to. No multiplication operations were performed (p. 15, 4th paragraph):

“The pH-dependent fluorescence intensity of FITC when excited at ~490 nm was used as described below to determine the steady-state pH of lysosomes. The relatively pH-insensitive fluorescence intensity of FITC when excited at ~440 nm was used to control for photobleaching during image acquisition. Multiple fields of cells were imaged for each condition, and the data was processed with Velocity v6.3. To convert fluorescence ratios to pH values, cells were sequentially subjected to isotonic K⁺ solutions (143 mM KCl, 5 mM glucose, 1 mM MgCl₂, 1 mM CaCl₂) at different pH values (pH 4.5 solution buffered with 20 mM acetic acid, pH 5.5, 6.5, and 7.5 solutions buffered with 20 mM MES), containing 10 μM nigericin and 5 μM monensin. Cells were imaged 5 min after the addition of each solution to obtain the 490 nm/440 nm fluorescence ratio corresponding to each pH standard. A calibration curve was prepared from a least squares fit of mean background-subtracted fluorescence ratios as a function of pH. The pH measurements were compared with an unpaired two-tailed T test to detect significant differences in measured pH.”

4. Experimental procedures for in vitro ATPase activity test / bafilomycin sensitivity (Fig 3A) is missing.

We have now added the following description to the methods sections:

“Enzyme-coupled ATPase activity assays were performed as described previously (Kornberg and Pricer, 1951; Vasanthakumar et al., 2019). Briefly, enzyme preparations were assayed in a 96-well plate with a total reaction volume of 160 μL. Purified V-ATPase was added to the ATPase assay reaction buffer (50 mM Tris pH 7.4, 3 mM

MgCl, 0.2 mM NADH disodium salt, 3.2 units pyruvate kinase, 8 units L-lactic dehydrogenase, 0.025% (w/v) GDN and the reaction was initiated with the addition of 1 mM ATP disodium salt and 1 mM phosphoenol pyruvic acid monopotassium salt. Either bafilomycin A1 in DMSO at 1 μ M or DMSO without bafilomycin was added to each well. Absorbance at 340 nm was monitored at 30°C to measure the signal from NADH and converted to concentration of NADH with a standard curve.

5. Authors drew functional hypotheses of mEAK7 from Figure 4, but ATP coupled release of mEAK-7 is unclear based on the provided information. It seems to be clear that its c-terminal domain plays role for binding to V-ATPase, but dominant 83% of particles bound intact in the presence of ATP, which is even more without ATP. 26% loosely bound population doesn't have strong structured features of mEAK-7 and so, it could be classification error, or ATP might induce structural instability or strengthen the binding for those misassembled particles. Authors may consider simple biochemical assay with/without ATP to show their interaction.

We apologize for not being sufficiently clear. In the absence of ATP it was impossible to find a 3D class that was completely free of mEAK-7 density and density for the mEAK-7's C-terminal α helix was always clear (Fig. 4A, left). In the presence of ATP, it became possible to produce a 3D class that lacked any detectable density for mEAK-7 and even the density for the C-terminal α helix was absent (Fig. 4A, left). The precise fraction of particle images going into each 3D class should likely not be interpreted as it is not a particularly precise measure of conformational distribution, as we have found previously (e.g. Di Trani *et al.*, 2022, Structure 6:129-138). To clarify what we meant we have revised the text (p. 8, 2nd paragraph):

“To determine the effect of ATP hydrolysis on mEAK-7 crosslinking of the rotor and the stator, V-ATPase was mixed with a ~20 \times molar excess of mEAK-7, ATP was added to 10 mM and mixed, and cryoEM grids were frozen within 5 s. Given the concentration of V-ATPase (20 mg/mL) and the enzyme's specific ATPase activity (2.9 ± 0.72 μ mol ATP/min/mg), these conditions ensure that the grids were frozen before the supply of ATP was consumed. In the absence of ATP it was impossible to find a 3D class in rotational State 2 that lacked density for mEAK-7 completely and density for mEAK-7's C-terminal α helix was always clear (Fig. 4A, *upper left*). In contrast, with ATP added a population of particles appeared with mEAK-7 entirely absent in rotational State 2 (Fig. 4A, *upper right*). In this state, even density for the C-terminal α helix is missing (Fig. 4A, *upper right, circled in orange*). Thus, it appears that rotation of the rotor, driven by ATP hydrolysis can result in the displacement of mEAK-7 from V-ATPase and disruption of the cross-link between the enzyme's rotor and stator. Despite several biochemical experiments, we were unable to demonstrate that mEAK-7 can be fully dissociated from V-ATPase by the addition of ATP to the complex, including experiments where we washed immobilized V-ATPase:mEAK-7 extensively with a buffer containing ATP before elution of V-ATPase. These experiments suggest that mEAK-7 can remain associated to V-ATPase, either by its TLDc domain or its C-terminal α helix, during ATP hydrolysis.”

Minor comments

1. 'V-ATPase binding by mEAK depends on its C-terminal α -helix and is ATP sensitive' - this title suggests that the C-terminal helix is binding determination domain, but a quarter of mEAK still can bind to the V-ATPase even in structure, which is not a negligible population. Toned-down paraphrasing of the title would describe the author's manuscript much clearly.

Here the reviewer is referring to the heading for the section on page 7. We have changed the section heading to:

“V-ATPase binding by mEAK involves its C-terminal α helix and is perturbed by ATP”

Full Revision

2. Increasing the box width of Fig3C would help much clear visualization

We have increased the box width as suggested.

3. Based on the illustration in the movie, there seems to be notable conformational changes induced by mEAK-7 binding. Authors may consider describing these structural differences.

The reviewer is absolutely correct. The conformational changes induced by mEAK-7 binding are shown in Supplementary Figure 6, panel D. We have added the following text (p. 6, 2nd paragraph):

“Binding of mEAK-7 to rotational State 2 causes numerous subtle conformational changes throughout the complex (Fig. S6D, Video 1), including widening of the subunit A-B interface where the C-terminal α helix binds, and a slight rotation of the central rotor in the direction of ATP hydrolysis.”

Question.

Is there any chance this mEAK-7 functions for assembly for V1-Vo-ATPase in certain cellular conditions?

We have thought about this possibility also. Although we do not have any evidence to support this idea, we are happy to speculate (p. 7, 2nd paragraph)

“mEAK-7 binding could even participate in V-ATPase assembly in some cellular contexts.”

Reviewer #3 (Significance (Required)):

This manuscript shows that mEAK-7, which possess the TLD domain, is a binding partner to V-ATPase whose binding is affected by ATP sensing in structural and biochemical analyses. It corresponds to the previous studies and suggest a model for regulation of V-ATPase in combination with diverse regulators. Researchers whose research field is related with ATP pump and network including the pump, pH-dependent regulation, and structural biology (using cryo-EM, especially) would be the one influenced by the reported findings. These comments are given by structural biologist, with experience in cell biology and biochemistry experiments to support the structural findings.

We agree with the above assessment but also believe that structural biologists will find our approach for identifying scarce binding partners of proteins valuable (as suggested by Reviewer #1).

June 13, 2022

RE: Life Science Alliance Manuscript #LSA-2022-01527R

Prof. John L Rubinstein
The Hospital for Sick Children
Molecular Structure and Function Program
686 Bay Street, Rm. 20-9705
Toronto, ON M6S 4H7
Canada

Dear Dr. Rubinstein,

Thank you for submitting your revised manuscript entitled "CryoEM of endogenous mammalian V-ATPase interacting with the TLDc protein mEAK-7". We would be happy to publish your paper in Life Science Alliance pending final revisions necessary to meet our formatting guidelines.

- please add a category for your manuscript to our system
- please add the Twitter handle of your host institute/organization as well as your own or/and one of the authors in our system
- please consult our manuscript preparation guidelines <https://www.life-science-alliance.org/manuscript-prep> and make sure your manuscript sections are in the correct order
- please use the [10 author names, et al.] format in your references (i.e. limit the author names to the first 10)
- please add the video legend and the supplementary figure legends to the main manuscript text
- please double check the callouts for Figure 5; you have callouts for Figure 5C, but these are not in the figure legend or the figure itself
- the 2 Supplementary Data files should be uploaded as Supplementary Tables, with titles

A. FINAL FILES:

B. MANUSCRIPT ORGANIZATION AND FORMATTING:

Sincerely,

Reviewer #1 (Comments to the Authors (Required)):

The modifications made have further strengthened the manuscript improving the clarity and providing further analysis. I believe this will be of interest to the field and will provide the basis for further studies into the V-ATPase and interacting partners. I do not think any further revisions are required.

Reviewer #2 (Comments to the Authors (Required)):

This study brings together cryo-EM, Mass spectrometry and some well considered molecular biology to discuss a new binding partner of the V-ATPase (mEAK-7) and map out its binding interactions. This combinational efforts provide sufficient progress to convince the reader with some interesting questions for future studies in the field.

I appreciate authors extra efforts to provide further clarity in the manuscript. Authors provide reasonable point by point response in an excellent manner.

I don't find any remaining issue in the manuscript.

June 21, 2022

RE: Life Science Alliance Manuscript #LSA-2022-01527RR

Prof. John L Rubinstein
The Hospital for Sick Children
Molecular Structure and Function Program
686 Bay Street, Rm. 20-9705
Toronto, ON M6S 4H7
Canada

Dear Dr. Rubinstein,

Thank you for submitting your Research Article entitled "CryoEM of endogenous mammalian V-ATPase interacting with the TLDc protein mEAK-7". It is a pleasure to let you know that your manuscript is now accepted for publication in Life Science Alliance. Congratulations on this interesting work.

DISTRIBUTION OF MATERIALS:

Again, congratulations on a very nice paper. I hope you found the review process to be constructive and are pleased with how the manuscript was handled editorially. We look forward to future exciting submissions from your lab.

Sincerely,
